# *Dioscorea* spp.: Comprehensive Review of Antioxidant Properties and Their Relation to Phytochemicals and Health Benefits

**DOI:** 10.3390/molecules27082530

**Published:** 2022-04-17

**Authors:** Aušra Adomėnienė, Petras Rimantas Venskutonis

**Affiliations:** Department of Food Science and Technology, Kaunas University of Technology, Radvilėnų˛ pl. 19, LT-50254 Kaunas, Lithuania; ausra.adomeniene@ktu.edu

**Keywords:** *Dioscorea*, antioxidant potential, radical-scavenging capacity, phytochemicals

## Abstract

*Dioscorea*, consisting of over 600 species, is the most important genus in the Dioscoreaceae family; however, the practically used plants, which are commonly called yam, are restricted to a remarkably smaller number of species. Numerous studies have reported the high nutritional value of yam, particularly as an alternative source of starch and some important micronutrients. Several *Dioscorea* species are widely used for various medicinal purposes as well. In many studies, the bioactivities and health benefits of *Dioscorea* extracts and other preparations have been related to the presence of phytochemicals, which possess antioxidant properties; they are related mainly to radical-scavenging capacity in chemical assays and positive effects on the endogenous antioxidant system in cell-based and in vivo assays. Considering the increasing number of publications on this topic and the absence of comprehensive and focused review papers on antioxidant potential, this article summarizes the results of studies on the antioxidant properties of *Dioscorea* spp. and their relation to phytochemicals and health benefits. A comprehensive survey of the published articles has revealed that the majority of studies have been performed with plant tubers (rhizomes, roots), while reports on leaves are rather scarce. In general, leaf extracts demonstrated stronger antioxidant potential than tuber preparations. This may be related to the differences in phytochemical composition: saponins, phenanthrenes and, for some pigment-rich species (purple yams), anthocyanins are important constituents in tubers, while phenolic acids and flavonoids are characteristic phytochemicals in the leaves. The review may assist in explaining ethnopharmacological knowledge on the health benefits of *Dioscorea* plants and their preparations; moreover, it may foster further studies of poorly investigated species, as well as their wider application in developing new functional foods and nutraceuticals.

## 1. Introduction

The role of fruits, vegetables and botanicals, as natural and sustainable sources of valuable nutrients and health-beneficial substances, has been very important for humankind since ancient times. Due to the vast biodiversity in the plant kingdom and the large number of poorly investigated plants, the trend in the search, evaluation and application of natural preparations—as well as the development of new products—remains an important issue in various industries, including foods, pharmaceuticals and cosmetics. Moreover, the importance of this topic is increasing in the era of functional foods, nutraceuticals and personalized nutrition. In addition, the shift in consumer preferences towards ‘naturalness’ has also played an important role in the popularity of plant origin preparations.

Dioscoreaceae, with nine genera and about 715 species, is considered one of the earliest families of the plant kingdom (Magnoliophyta) [1]. In terms of uses, *Dioscorea,* consisting of over 600 species, is the most important genus in this family. The plants are perennial herbaceous and dicotyledonous, and bloom with faint flowers and have large roots and/or rhizomes. *Dioscorea* plants are common in tropical and subtropical regions; however, many of them have successfully adapted to different habitats and, nowadays, are common in other climatic regions as well. Regardless, a large number of existing species—practically used plants, which are commonly called yam—are restricted to a smaller number of species; among them are *D. batatas*, *D. japonica*, *D. bulbifera*, *D. opposita*, *D. tokoro*, *D. nipponica* and *D. alata* [2]. For instance, *D. bulbifera* L. (air potato) is native in Asia, North Florida [3], and Georgia (www.gaeppc.org/list/ [4], accessed on 11 November 2021); *D. alata* L. (water yam, winged/white yam) originates from Southeast Asia [5] but is also found in Africa (FAOSTAT, Database 2021 [6], accessed on 11 November 2021), Georgia (www.gaeppc.org/list/ [4], accessed on 11 November 2021), and America (www.eddmaps.org [7], accessed on 12 November 2021); *D. oppositifolia*, syn. *D. opposita*, *D. batata*, *D. polystachya* Turcz. (Chinese yam), is native to Asia and is also found in eastern North America.

It should be noted that the occurrence of invasive plant species, albeit beneficial ones, poses a threat to native species and habitats. For example, *D. oppositifolia* (syn. *D. polystachya*) is recognized in North America (www.iucngish.org/gisd/, accessed on 22 November 2021, Global Invasive Species Database, 2021 [8]) as a forest or agricultural weed, which can compete with the native species, especially in coastal habitats; while widespread in Africa, *D. alata* is one of the most valued foods. In general, due to the nutritional and therapeutic benefits and economic income, the contribution of yam to food security is significant in some regions, especially in sub-Saharan Africa [9]. *Dioscorea* spp. are also known for its association with low-cost food culture, traditional Chinese medicine, modern western medicine, and the pharmaceutical industry. The peculiarities of the diet and traditions of this genus of plant species in many cultures are closely related to the historical and religious aspects of life (FAOSTAT, Database, 2021 [6]), which began much earlier than scientific research. It should be noted that different yam species accumulate a large variety of biologically active compounds and, therefore, various anatomical parts of the plants have been used for different purposes. For instance, *D alata*, *D. japonica* Thunb., and the rhizomes of *D. schimperiana* [10,11,12] are valued as a source of starch in the development of flour, pasta, and desserts.

Traditional medicine mentions such species as *D. bulbifera* in the treatment of goiter, skin infections, and oncological diseases [13]; *D. nipponica* in the treatment of arthritis, cough, asthma, and circulatory disorders [14]; *D. villosa* (wild yam) in the reduction of intestinal disorders [15]; and *D. birmanica* in the treatment of chronic diseases, whose development is associated with long-term oxidative stress [16]. Food supplements used in modern medicine as natural alternatives to hormone replacement therapy, in the form of powders, capsules, and herbal extracts, are made from the roots of *D. villosa*; a traditional herbal preparation made from *D. nipponica* root is used to regulate the immune system and reduce pain and inflammation [14], while the saponins extracted from *D. septembola* are used in China to treat gout [17].

In addition to the well-known species with high nutritional, economic and medicinal value, it is worth mentioning the other less-known but valuable species that can enrich the diet and improve health, namely: *D. birmanica* Prain and Burkill [16]; *D. hamiltonii* Hook. f syn. *D. persimilis* [18,19]; *D. hispida* [20]; *D. pubera* Blume; *D. wallichii* Hook [21], and others. Some species are poisonous but have healing properties, e.g., *D. hispida*, which is a promising alternative to conventional chemotherapy [22,23].

Secondary metabolites identified in *Dioscorea* spp. are often used in physiological and molecular studies. The well-known natural steroid saponin diosgenin has been used in the pharmaceutical industry as a chemical model in the development and/or complete synthesis of hormonal drugs. This metabolite is also reported to reduce oxidative stress damage, may protect the myocardium from ischemia-induced injury [24], and modulates the intestinal microbiota [24,25]. Dioscin and gracillin are also very important metabolites, demonstrating anticancer [26,27] and antioxidant effects. Trillin, found in *D. nipponica* [28], *D. oppositifolia*, and *D. hamiltonii* [29], is a pharmacologically viable steroid saponin with a broad spectrum of physiological effects in the cells, e.g., increasing superoxide dismutase (SOD) activity, and lowering lipid peroxidation activity and the oxidative stress response [28].

From nutritional and economic points of view, carbohydrates are likely the most important constituents of yam tubers, which are used in the development of foods as bulk ingredients; however, their content in the leaves is rather small. On the other hand, the leaves, as a rule, accumulate larger amounts of bioactive phytochemicals, e.g., polyphenolic antioxidants, including phenolic acids (chlorogenic, syringic, vanillic, *p*-hydroxybenzoic, *p*-coumaric), flavonoids (quercetin, rutin, kaempferol, catechin, anthocyanins), etc. Aromatic phenanthrene derivatives, demonstrating antifungal [30] and antioxidant effects [31,32], have also been reported in different species, e.g., *D. batatas* [30,32], *D. rotundata* [30], and *D. communis* [31].

As may be judged from the records present in various databases (Table 1), *Dioscorea* plants have been widely studied, and the published results have been reviewed in numerous articles.

The topic of *Dioscorea* antioxidants has also been the focus of many studies; however, the reviews on this topic are less abundant and fragmental.

Obidiegwu et al. [9] published the most comprehensive review on the nutritional and therapeutic potential of the *Dioscorea* genus. This review covers the nutritional value (starch, fiber, protein, fat, minerals), the profile of bioactive compounds, the therapeutic potential and health benefits (anti-inflammatory, anticancer, anti-diabetic, anti-obesity and anti-hypercholesterolaemic and antimicrobial activities), the management of degenerative disease, menopausal symptoms, and its use as a pharmaceutical excipient. Regarding bioactive compounds, the review includes separate sections on steroidal saponin, dioscorin, alkaloids, flavonoids, phenolic acids and others, while the antioxidant properties of yam are discussed in a very short section. In addition, it refers to seven publications from 2019, while in the period of 2019–2021, fifty-four new articles related to *Dioscorea* antioxidants have been published.

Padhan and Panda [33], in their review on underutilized and neglected yams, emphasized their potential for improving nutritional security and providing health benefits. The authors focused their article on the ethnobotany of *Dioscorea* spp., which are consumed by tribal people, and briefly highlighted their nutritional, anti-nutritional, and pharmacological properties. Kumar et al. [34] reviewed the traditional uses and ethnopharmacological potential of wild edible *Dioscorea* tubers by the local people of the Similipal Biosphere Reserve in India, whereas George et al. [35] highlighted the importance of African yam bean. Salehi et al. [36] reviewed *Dioscorea* plants as rich in nutrients and pharmacologically valuable constituents; however, their antioxidant activities were summarized in a very short section.

Some reviews were focused on individual *Dioscorea* species and important groups of nutrients and bioactives. For instance, Bhujbal et al. [37] reviewed the pharmacological activity of *D. floribunda*, and Semwal et al. [38] focused on the traditional uses, bioactive compounds and biological activities of *D. deltoidea* Wall. ex Griseb. Huang et al. [39] focused their paper on the isolation, structure and bioactivities of yam mucilage polysaccharides and included a short section on antioxidant properties, while Anwar et al. [40] included yam tubers in their review on water-soluble non-starch polysaccharides of root and tuber crops. Zhang et al. [41] focused their review on the development of proteins and peptides from *Dioscorea* tubers with therapeutic potential. Petropoulos et al. [42] reviewed the antioxidant properties and health benefits of colored root vegetables, which included a section on yam pigments as well. Finally, Yang et al. [43] reviewed the pharmacological activities of dioscin, and Toth et al. [44] reviewed those of phenanthrenes.

The survey of published review articles reveals that a comprehensive review on *Dioscorea* antioxidant properties has not been published until now, while the pool of data on this topic is quite large. Therefore, the main task of this review was to collect the available results on *Dioscorea* antioxidant potential, the presence of the main chemical groups of antioxidants, and their relations to health benefits. It should be noted that ethnopharmacological properties and medicinal uses are not covered in this review unless they are closely related to the antioxidant activities.

## 2. In Vitro Antioxidant Potential and Total Amounts of Phytochemicals

### 2.1. Evaluation of Crude Extracts and Their Fractions

The antioxidant properties of *Dioscorea* spp. have been studied using various in vitro chemical methods, cell cultures and in vivo assays, mainly using rat and mouse models. The results of the in vitro chemical assays are summarized in Table 2.

It can be observed that antioxidant properties have been determined using several chemical assays and expressed in various units, e.g., the equivalent amount of the reference antioxidant (trolox, ascorbic acid, catechin, α-tocopherol); the fresh (pfw) and dry weight (pdw) of plant material and/or extract (edw); the effective inhibitory concentration (IC_50_, in some cases abbreviated as EC_50_) values of the extract or its solution; or the percentage inhibition at the applied extract concentrations. In the latter case, the results are very difficult to compare because it is necessary to know the procedure of the sample preparation for the assays, particularly their concentrations in the solutions. Therefore, if the reader would like to know more details, it is suggested to download the full article. The values expressed in the reference antioxidant equivalents in pdw/pfw and/or edw are the most convenient; the former indicates the antioxidant potential of the whole plant material used in the assay, while the latter is directly related to the antioxidant properties of the isolated extracts and fractions.

This section also includes the results reported for the total phenolic content (TPC), total flavonoid content (TFC), total monomeric anthocyanins (TAC) and total saponins (TS) (Table 3). These values are determined mainly using spectrophotometric methods, and—depending on the plant material (species, cultivar), its composition, processing and extractions methods—may be rather indicative. It can be clearly observed from the recently reported values for MeOH extracts of *D. bulbifera* stem tubers (Table 3) that TPC and TFC values were determined to be remarkably lower than the total flavonols, which belong to one of the flavonoid subclasses [92]. Adeniran and Sonibare [67] reported somewhat similar findings; TFC values were remarkably higher than TPC ones in the four analyzed *Dioscorea* samples. TPC measurement with the Folin–Ciocalteu reagent is based on a single-electron-transfer reaction, which can proceed within the reaction system and present various nonphenolic compounds as well. In addition, the total content values in many publications are presented together with radical scavenging and other antioxidant capacity assays; therefore, it is reasonable to review all these values in one chapter. Nevertheless, the values of total phytochemical content are summarized separately in Table 3.

Several studies compared the antioxidant potential of different *Dioscorea* spp. and cultivars of the same species and demonstrated that the differences may be quite remarkable. For instance, Padhan et al. [101] reported the flavonoid, total antioxidant capacity and in vitro antioxidant activity of eight wild (*D. oppositifolia*, *D. hamiltonii*, *D. bulbifera*, *D. pubera*, *D. pentaphylla*, *D. wallichii*, *D. glabra*, *D. hispida*) and one cultivated (*D. alata*) yam tuber from Koraput (India): TPC, TFC, and total antioxidant capacity ranged from 2.19 to 9.62 mg GAE/g pdw, 0.62–0.85 mg QE/g pdw and 1.63–5.59%, respectively; meanwhile, the IC_50_ values were 77.9–1164, 101.2–1031.6, 27.0–1022.6 and 47.1–690 µg/mL for DPPH^•^, ABTS^•+^, ^•^O_2_^−^, and ^•^NO scavenging capacity, respectively. The TPC and TFC values of *D. deltoidea*, *D. prazeri*, *D. bulbifera*, *D. pentaphylla*, *D. esculenta*, and six cultivars of *D. alata* were in the ranges of 115.31–1628.50 mg/100 g pdw and 17.57–148.46 mg/100 g pdw, with D. prazeri being the richest source of polyphenolics [102]. Bhandari and Kawabata [96] reported that the TPC of *D. bulbifera*, *D. versicolor*, *D. deltoidei*, and *D. triphylla* from Nepal ranged from 13 ± 1 to 166 ± 10 mg/100 g fpw, while the differences in antioxidant characteristics were less remarkable, particularly in chelating ferrous ion, reducing power, and total antioxidant capacity. Jimenez-Montero and Silvera [103] compared six varieties of *D. bulbifera* (air potato) from Panama and found that the variations in ascorbic acid, antioxidant capacity, and TPC were not remarkable (6.0–8.1 mg/100 g, IC_50_ 7.85–8.87, and 175–201 mg GAE/100 g, respectively). However, the cultivars of the same species may differ significantly. For instance, the TPC, TFC, and TAC values of the flour of the Kulonprogo cultivar of *D. alata*, determined by extraction with acid and with 5% acetic acid water, were higher than those of the Malang cultivar [49]. Alcohol and ester extracts of yellow fresh-cut yam (*D. opposita*) exhibited higher ORAC and DPPH^•^ scavenging capacity than white fresh-cut yam [69].

The majority of studies used yam; however, some of them were performed with different anatomical parts of *Dioscorea* as well, including the flesh, peels and leaves. Thus, Alsawalha et al. [104] reported that MeOH extract from *D. villosa* leaf powder demonstrated strong antioxidant capacity in a FRAP and DPPH^•^ scavenging assay in a dose-dependent manner; at a concentration of 0.1 mg/mL, the inhibition of DPPH^•^ was >90%, while the optical density in the FRAP assay was 1.3 (2 for ascorbic acid). Mondal et al. [105] compared MeOH and EA extracts of D. pentaphylla leaves and found that TPC, DPPH^•^ and ^•^NO scavenging was higher for the former, while the latter had almost 4-fold higher TFC and TAC.

Yams contain various nutrients and phytochemicals, which differ in their chemical structures, molecular weight, polarity and other characteristics. Some studies fractionated yam materials and extracts using different polarity solvents. Duan et al. [60] reported only slight differences between n-BuOH and EA extracts of *D. batatas*, which were evaluated using different in vitro antioxidant assays. In the case of a thermally treated product, EA extract was more effective in chelating ferrous ion and scavenging ^•^NO, while n-BuOH extract better inhibited linoleic acid peroxidation [59]. Duan et al. [56] compared yam (*D. batatas*) extracts isolated with 70% MeOH, 70% EtOH and CF–MeOH (2:1, *v/v*). MeOH extract was most effective at chelating ferrous ion, ^•^NO scavenging and the β-carotene bleaching assay, while CF–MeOH extract better inhibited linoleic acid peroxidation. Ghosh et al. [64] extracted the bulbs of *D. bulbifera* with cold 70% EtOH, and sequentially with PE, EA and MeOH; the latter extract contained higher TPC, while the EA fraction was richer in TFC (Table 3). The antioxidant properties of EA and alcoholic extracts were stronger compared with the PE extract. Park et al. [89] extracted bulbils of Danma (*D. japonica*) and Jangma (*D. batatas*) with MeOH, and the crude extracts were re-extracted with CF, EA, n-BuOH, and water. Xia et al. [106] compared antioxidant properties of PE, EA and n-BuOH fractions of *D. nipponica*, which were obtained from a dried 60% EtOH extract. The fractions inhibited DPPH^•^, ^•^O_2_^−^, and ^•^OH in a concentration range of 2.5–12.5 mg/mL; however, the dependence between radical inhibition percentage and extract concentration was far from linear. Adeniran et al. [107] successively extracted the peel and flesh of *D. dumetorum*, *D. hirtiflora*, and *D. bulbifera* with the increasing polarity solvents HX, DCM, EA, and MeOH. The MeOH extract of *D. hirtiflora* flesh showed the highest radical-scavenging capacity, while TPC and TFC were higher in the EA extract of *D. bulbifera* flesh. Among *D. quinqueloba* extracts isolated with water, MeOH, EtOH, and EA, the EtOH extract was the strongest DPPH^•^ scavenger, whereas EA extract better scavenged ABTS^•+^; both values increased with the addition of water-soluble chitosan [97]. Chen et al. [53] fractionated water solution of dried crude EtOH extract from Taiwan yams (*D. alata*) via sequential extraction with HX, CF, EA, and n-BuOH. Based on the IC_50_ values, the peel extracts better scavenged DPPH^•^ than the flesh products. The EA fraction of the peel of Tainung yam was a particularly strong antioxidant; its EC_50_ value was only 14.5 µg/mL, which was lower than that of ascorbic acid (21.4 µg/mL). In addition, the authors applied pressurized solvents and revealed that hot pressurized EtOH was superior to hot pressurized water in extracting the DPPH^•^ scavenging compounds [53].

Only one study was found on the use of biotechnological methods for obtaining antioxidants from yams. Jirakiattikul et al. [108] applied in vitro propagation of *D. birmanica* shoots; however, the TPC and DPPH^•^ scavenging values were remarkably higher in the mother rhizomes than in the in vitro produced shoots (259.67 ± 7.34 vs. 44.24 ± 8.47 mg GAE/g and 11.42 ± 3.28 vs. 53.67 ± 4.16 EC_50_ µg/mL dry extract, respectively).

### 2.2. Evaluation of Polysaccharides and Some Other Compounds

Several studies reported antioxidant properties in yam polysaccharides. The pre-purified (dialyzed) polysaccharide of air yam (*D. bulbifera*) was approx. a 4-fold stronger antioxidant in ABTS^•+^, according to a decolorization assay, than the crude (non-dialyzed) polysaccharide; in contrast, in DPPH^•^ and FRAP assays, the latter showed a several-times stronger antioxidant capacity [65]. It can be noted that the reported results are rather strange, because the mechanisms of all the radical-scavenging reactions were based on electron transfer (hydrogen atom transfer may proceed with DPPH^•^ as well). Most likely, extended studies using more specific assays would be required to more precisely clarify the mechanisms of action of yam polysaccharides in the redox systems. Zhang et al. [105] precipitated yam polysaccharide from the centrifugate of *D. opposita* homogenate with 75% EtOH, and afterwards, degraded it into three low MW derivatives; these were strong antioxidants against ^•^OH (IC_50_ = 187, 82 and 55 µg/mL) and ^•^O_2_^−^ (IC_50_ was 241, 138 and 125 µg/mL), while the activity of raw polysaccharide was lower. The polysaccharide partially prevented the cyclophosphamide-induced damage, and its effect was probably due to high anti-lipid peroxidation activity and radical-scavenging effects. A previously unknown water-soluble polysaccharide was isolated from the roots of a well-known edible and medicinal plant in China, *D. opposite*. Its antioxidant capacity IC_50_ values against DPPH^•^ and PTIO^•^ were 2.1 ± 0.1 and 1.6 ± 0.1 mg/mL, respectively [109]. Zhao et al. [72] compared ultrasound-assisted, cold, warm, and hot water extraction for the isolation of polysaccharides, and determined that cold water gave the highest uronic acid yield; its DPPH^•^ scavenging, α-glucosidase and α-amylase inhibition activities were the best. The MW of polysaccharides recovered using ultrasound-assisted extraction was small, while its reducing power and FRAP were the best. The extracts of tuber pulp from the three *D. alata* taxa had significant anti-Fenton reaction activity, which was related to the polysaccharide mucilage [110]. Most recently, the antioxidant activity of mucilage isolated from *D. opposita* was evaluated using DPPH^•^-scavenging and FRAP assays; at a 5.0 mg/mL concentration, it inhibited 68.57% of radicals, while FRAP was remarkably lower compared to ascorbic acid [111]. The ^•^OH-scavenging capacity of diosgenin obtained from *D. composita* by direct hydrolysis-extraction, under biphasic reaction conditions (1.5% yield), was higher than that of ascorbic acid when the concentration of the scavenger was higher than 2.5 × 10^−3^ mmol/L [112]. Various solvents have been used for the extraction and fractionation of antioxidants from *Dioscorea*. Kwon et al. [45] determined antioxidant, antithrombin, and antimicrobial activities of MeOH extracts prepared from two *D. batatas* cultivars, two *D. alata*, one *D. bulbifera*, and one *D. nipponica*. Ramos-Escudero et al. [113] reported the TAC, TPC, and antioxidant activity of the pigments extracted from *D. trifida.*

### 2.3. Evaluation of Proteins and Their Hydrolysis Products

Several studies reported antioxidant and other bioactivities of yam mucilage, proteins and their hydrolysates. Hou et al. [114] purified the storage protein of yam (*D. batatas*) tuber dioscorin (32 kDa), which scavenged DPPH^•^ (8–46%) at 5.97–47.80 nmol; dioscorin decreased the intensities of the electron spin resonance (EPR) signals used for DPPH^•^ and ^•^OH detection in a dose-dependent manner. Hou et al. [115] extracted and partially purified mucilage from a *D. batatas* tuber: its IC_50_ in DPPH^•^, ^•^OH and anti-lipid peroxidation and anti-human low-density lipoprotein peroxidation tests were 0.86 mg/mL, 22 µg/mL and 145.46 µg/mL, respectively, using BHT, GSH, or ascorbic acid for comparisons. The intensities of the EPR signals decreased with increasing yam mucilage amounts (IC_50_ was 1.62 mg/mL). The mucilage also inhibited the angiotensin-converting enzyme (ACE) [116]. Nagai and co-authors [81,85,117,118,119,120] studied proteins isolated from *D. opposita* tuber mucilage tororo. Firstly, they obtained viscous water extract containing 280 mg/mL of protein, which demonstrated the main bands determined by SDS-PAGE at MW ~31 kDa (reducing conditions) or ~33 kDa (non-reducing conditions), and scavenged ^•^O_2_^−^ (84.1 ± 6.57%), ^•^OH (79.4 ± 6.42%), and DPPH^•^ (38.2 ± 3.14%) [61]. Afterwards they treated the extract with protease or mannanase and noted that the viscosity highly depends on the interactions between certain soluble proteins with mannan [121]. They purified a soluble viscous protein, dioscorin (~200 kDa), from the mucilage using a chromatographic procedure [111] and measured its antioxidant properties. The protein effectively scavenged ^•^OH (IC_50_ = 195.1 μg/mL) and ^•^O_2_^−^ (IC_50_ = 92.7 μg/mL) [120]. Autolysate and trypsin hydrolysate from yam (*D. opposita*) ichyoimo tubers exhibited extremely high antioxidant and ACE-inhibitory activities [119]. The lyophilized powder of enzymatic hydrolysates from mucilage tororo effectively scavenged ^•^O_2_^−^ and ^•^OH, particularly when trypsin was used for hydrolysis [118]. The hydrolysates of a yam tsukuneimo tuber mucilage tororo at a 100 mg/mL concentration significantly prolonged the induction period of linoleic acid auto-oxidation (similar to 5 mM ascorbic acid), and scavenged ^•^O_2_^−^, ^•^OH, and DPPH^•^ [85]. Liu et al. [122] also purified dioscorin from *D. alata* and *D. batatas* using DE-52 ion-exchange chromatography and reported antioxidant properties in the protein and its hydrolysates using various assays. Oh and Lim [123] isolated glycoprotein from *D. batatas*, which has been traditionally used as a folk medicine and health food in Korea; it had an optimal radical-scavenging capacity at an acidic and neutral pH and at temperatures up to 85 °C, while in the presence of Ca^2+^, Mn^2+^, Mg^2+^, and EDTA, its activity reduced.

### 2.4. Effects of Processing and Various Treatments

Processing techniques such as blanching, thermal treatment, soaking, oven-drying, and freeze-drying had different effects on the content of phenolics, DPPH^•^ scavenging and polyphenol oxidase activity in *D. esculenta* and *D. bulbifera* tuber flour [124]. Chung et al. [88] reported the effects of blanching, drying and extraction with 50% EtOH, hot water, and water on the antioxidant activities of Taiwanese yam (*D. alata*). The peel exhibited much higher antioxidant activities. Blanching by immersing the peel in 85 °C water for 30 s caused significant reduction in the antioxidant activities of all peel extracts. Generally speaking, freeze-dried peel maintained higher antioxidant activities than hot air-dried peel. More recently, Santos et al. [90] reported that the freeze-dried purple yam (*D. trifida*) powder had higher antioxidant capacity and higher TPC and TAC values, compared with the products dried using a refractance window at 70, 80, and 90 °C for 40 min. Compared with the native starch, the antioxidant activity of the starch–palmitic acid complex, prepared using the reaction of different concentrations (0.1–5.0%) of palmitic acid and yam (*D. opposita*) starch, significantly increased [125]. *D. bulbifera* tuber extract was used in the synthesis of novel platinum–palladium bimetallic nanoparticles along with individual platinum and palladium nanoparticles. Bimetallic nanoparticles demonstrated enhanced DPPH^•^, ^•^O_2_^−^, ^•^NO, and ^•^OH scavenging capacity and more pronounced effects on the HeLa cancer cell death [126]. There was no particle size influence on the content of phenolics (0.27–2.82% dw) or anthocyanin (2.25–15.27 mg/100 g dw) in light purple, purple and dark purple cultivars, or carotene (23.75–132.12 mg/100 g dw) in yellow and orange cultivars of *D. alata* [99]. Adedayo et al. [47] reported the effects of processing on the TPC, TFC and antioxidant characteristics of white (*D. rotundata*) and water yam (*D. alata*) flour (Table 3). Jaleel et al. [127] determined that paclobutrazol (PBZ) treatments had a profound effect on antioxidant metabolism and caused an enhancement in both nonenzymatic and enzymatic antioxidant potentials under treatments in white yam (*D. rotundata*). Djeukeu et al. [11] used 10–60% of the dried yam (*D. schimperiana*) flour produced by the traditional and modified process (boiled for polyphenol oxidase inhibition before drying) methods in pasta; a yam ingredient increased TPC, ABTS^•+^ scavenging, FRAP, and RSA values up to 2.9, 3.80, 14.48, and 2.21-fold; conversely in the case of the modified process, all the values were higher than for the traditional method. The content of total soluble phenolics in the roots of minimally processed yam (commercial *Dioscorea* spp.) did not change significantly during 14 days of storage at 5 °C (16.9–17.8 mg/g FW) or at 10 °C (16.7–17.5 mg/g FW) [128]. Lu et al. [98] reported an increase in the content of polyphenols, flavonoids, and anthocyanins in the tuber flesh and peel of five commercial yams using arbuscular mycorrhizal fungi treatment. Ratnaningsih et al. [49] compared immersion in water, Na-bisulfite, and ascorbic acid as browning inhibitors of *D. alata* flour, and found that the treatment provided better TAC, TPC and TFC values.

### 2.5. Application of Dioscorea spp. Ingredients for Increasing Product Antioxidant Properties

Yam flour has been used in the formulations of various foods. Adeloye et al. [10] reported that the addition of water yam (*D. alata*) flour in the formula of dough at 10–40%, together with plantain and bitter leaves, increased the TPC from 1.26 to 4.44 mg GAE/g, the radical-scavenging capacity from 14.73 to 48.66 mg/g, and the FRAP from 8.57 to 42.52 mg AAE/g, in a dose-dependent manner. The substitution of up to 40% of wheat flour in bread formulation with purple yam (*D. alata*) flour, containing 220 mg GAE of TPC, 640 mg AAE DPPH scavenging capacity, and 16 mg CGE of TAC in 100 g, showed liking scores in all bread sensory attributes similar to those of the control [129]. The addition of 2–8% *D. japonica* powder—containing 11.16% protein, 0.73% fat, and 76.39% carbohydrate—into red bean sweet jelly desert ‘Yangaeng’ increased the TPC from 14.58 to 71.54 mg GAE/100 g, and DPPH^•^ scavenging from 5.84 to 63.98%, in a dose-dependent manner [12]. Extrusion processing increased the TBARS inhibition ability for all yams (*D. alata* var. Tai-nun and var. Ta-shan, D. doryophora var. Hang-chun and commercial yam flour) and generally had no negative impact on DPPH^•^, ABTS^•+^ scavenging, ORAC, and Fe^2+^ chelating for most yams in corn–yam extrudates [51]. Extruded yam flour (*D. alata*) containing 61.20 ± 0.02 mg GAE/g phenolics was used in the formulation of instant drinks [130]. Yams (*D. alata*) added at 5%, 10%, and 15% to Chinese sausages caused lower thiobarbituric acid (TBA) values during 21 days of storage [131]. Li et al. [132] tested up to 25% of Chinese yam (*D. opposita*) flour in bread formula. The content of flavonoids increased in the bread from 0.162 to 0.516 mg RE/g, and allantoin from 0.05 to 1.063 mg/g, while the IC_50_ value of antioxidant capacity decreased from 124.14 to 101.36 mg/mL. On the other hand, it should be noted that the TPC values increased less significantly, from 0.467 to 0.491 mg GAE/g, which is in some degree of disagreement with the total flavonoids, which belong to the group of phenolics.

## 3. In Vitro Assays in Cell Cultures and In Vivo Assays with Animals

Quite a few studies tested Dioscorea preparations in cell cultures and in vivo assays with animals, mainly rats and mice. The results of these studies are summarized in Table 4.

The effects of yam extracts, purified fractions, and individual compounds on various biomarkers were reported, including important factors of endogenous antioxidant defense in the cells, namely superoxide dismutase (SOD), glutathione peroxidase (GPx), catalase (CAT) and reduced glutathione (GSH). This table also includes brief information on chemical assays of enzyme inhibition, which are related to various physiological effects. It should be noted that many authors related the beneficial effects of *Dioscorea* preparations in cell and animal assays with their antioxidant properties—which were determined using the chemical assays (Table 2)—and the presence of various classes of phytochemicals (Table 3).

The extracts isolated with different polarity solvents were tested in various cell assays. EtOH extraction of dried, ground *D. birmanica* (locally called Hua-Khao Yen-Neua) rhizomes yielded 10.56% of the extract, with TPC and TFC values of 170.85 ± 3.02 mg GAE/g and 132.55 ± 3.59 mg CE/g, respectively [16]. The authors suggested that the identified phytochemicals from the extract, which scavenged radicals and moderately chelated metals, may be responsible for various bioactivities in the cells. Water fraction isolated by reflux-extraction of dried *D. cirrhosa* powder, and afterwards separated by AB-8 macroporous resin with water, 60% and 95% EtOH demonstrated numerous bioactivities in cells as well [148]. The extracts of aerial bulblets of *D. japonica* inhibited the induction of inflammatory mediators and the expression of iNOS and COX-2 in the cells [149].

Among the five products separated from the purple Chinese yam (*D. alata*) extracts, the EA fraction of flesh and peel exerted the best antioxidant activity and exhibited excellent antiglycation ability by suppressing different stages of the glycation cascade; they also exhibited cytoprotective effects on methylglyoxal-induced oxidative damage in HepG2 cells [52]. Steroidal glycosides, spirostans, of *D. villosa* (zingiberensis saponin I, dioscin, deltonin, and progenin III) were found to be cytotoxic, whereas, furostans (*huangjiangsu A, pseudoprotodioscin, methyl protobioside, protodioscin, and protodeltonin*) were non-cytotoxic. While the treatment of HepG2 cells with compounds prior to H_2_O_2_ exposure effectively increased cell viability in a concentration-dependent manner, some of them increased GSH level and decreased intracellular ROS generation [150]. 2,7-Dihydroxy-4,6-dimethoxy phenanthrene, which was isolated from yam (*D. batatas*) peel extract with 95% EtOH, was found to be a strong antioxidant and anti-inflammatory agent in bio-guided fractionation assays with HepG2-ARE cells [136]. MeOH extract of *D. bulbifera* leaves had significantly higher antioxidant activity than EA and HX extracts in DPPH^•^, FRAP, and ABTS^•+^ assays, and demonstrated a pronounced cytotoxic effect in MDA-MB-231 and MCF-7 cell lines [63]. Ren et al. [151] isolated two new norsesquiterpenoids, dioscopposin A and dioscopposin B, as well as 21 known compounds from the stems and leaves of *D. oppositifolia*, and evaluated their estrogenic activity using MCF-7 cells; eight of these compounds inhibited cell proliferation. Antioxidant capacity was correlated with phenolic content in the cytoprotective effect on HUVEC viability assay, and *D. glabra* extract was more active than *D. alata* against H_2_O_2_-induced oxidative stress [152].

The in vivo studies were focused on the various physiological effects, and some of them reported beneficial effects of *Dioscorea* preparations in diabetic animals. Potential effects of ethanol extract isolated from a traditional Chinese medicine, *D. zingiberensis*, against testicular tissue damage via activation of the nuclear factor erythroid 2 (Nrf2) were demonstrated in a diabetic mouse model [153]. A polysaccharide, which was isolated from *D. opposita* roots with acidified water-extraction following precipitation with EtOH and chromatographic purification, increased SOD activity in alloxan-induced diabetic rats and mouse models at higher doses, and had strong hypoglycemic activity [154]. Fermentable water-soluble polysaccharides of purple and yellow water yam (*D. alata*) produced short-chain fatty acids, and also demonstrated hypoglycemic effects in rats with alloxan-induced glucose levels [134]. Treatment with crude yam (*D. batatas*) powder, water extract and allantoin effectively ameliorated antioxidant stress in streptozotocin (STZ)-induced diabetic rats by decreasing MDA formation and increasing SOD/GPx activities [140]. The administration of yam (*D. batatas*) rhizome aqueous extract and allantoin at high doses had numerous health benefits in the in vivo studies with high-fat-diet (HFD)/STZ-induced diabetic mice [137]. It is worth mentioning that the traditional Chinese medicine Rehmannia Six Formula for treating diabetes mellitus includes *D. opposita*, *D. alata*, and *D. batatas* [155].

Many in vivo studies reported antioxidant effects and their beneficial contribution to regulating inflammatory events and protecting against damage to important biomolecules through excessive formation of ROS. Several studies reported antioxidant defense-related benefits for gastrointestinal health. Ethanol extracts of *D. batatas* peel had various effects on human colonic epithelial HCT116 cells, including a reduction in the level of ROS in H_2_O_2_-challenged cells. Oral supplementation inhibited colon inflammation through the upregulation of antioxidant enzymes and subsequent downregulation of inflammatory proteins [156]. *D. batatas* flesh and peel extracts, obtained with water or 60% or 95% EtOH, were tested in vivo; pre-treatment with 60% EtOH flesh extract before ulcer induction significantly decreased the expression of biomarkers of oxidative stress [138]. Park et al. [18] also reported positive effects of crude mucin and saponins from the *Dioscorea* rhizome on rats induced with gastric ulcers using alcohol [18]. In the studies with gastric ulcer rats, the administration of crude Dioscorea rhizoma extract activated the antioxidant enzyme CAT and increased the levels of SOD and GP_X_, while the production of MDA, congestion, and hemorrhage of tissue decreased [157]. An anthocyanin-rich fraction with five identified compounds isolated from *D. alata* produced potent anti-inflammatory effects in the mouse model of inflammatory bowel disease (IBD) at 80 mg/kg bw; anthocyanins may reduce intestinal permeability, in part, by preserving the ultrastructural integrity of the epithelial mucosa [17]. Yam (*D. batatas*) increased erythrocyte levels of glutathione, GPx, and CAT in azoxymethane (AOM)-induced colon carcinogenesis in male F344 rats; in addition, the colonic mucosal gene expression of SOD and GPx were up-regulated, and the expression of inflammatory mediators was suppressed [158].

The excessive formation of ROS is also related with the initial phases of carcinogenesis. Yellow pigment from freshly cut yam (*D. opposita*) exhibited a protective effect against ^•^OH-induced DNA damage, which was attributed to the high ^•^OH-scavenging activity exerted by the pigment [70]. In addition, Liu et al. [75] determined that all the peel extracts of *D. opposita* had a better effect in the ROS scavenging assay and a stronger antitumor property in the mouse models than in flesh. *D. bulbifera* polysaccharides attenuated cyclophosphamide-heightened oxidative stress in mice; therefore, combination use can potentially enhance the cardiotoxin (CTX) anti-tumor effect and can attenuate CTX-induced immunosuppression and oxidative stress in U14 cervical-tumor-bearing mice [159].

Several studies focused on the evaluation of *Dioscorea* preparations as cardioprotective agents. Thus, total saponins from *D. nipponica*, *D. panthaica*, and *D. zingiberensis* lowered MDA levels and increased SOD, CAT, GPx, and T-AOC activities in rats; moreover, they demonstrated cardioprotective effects, as was determined by the less severe histological damage [146]. Isoproterenol (ISO)-induced rats pretreated with a flavonoid-rich fraction of *D. bulbifera* (150 mg/kg) exhibited ameliorated lipid peroxidation and, thereby, enhanced antioxidant status, as evidenced by the increase in the GSH content and the activity of antioxidant enzymes, which also suggest a cardioprotective effect [160]. Feng et al. [24] demonstrated that diosgenins (L, M, H) were the main metabolites produced by rat intestinal microflora from *D. nipponica*; moreover, when administered orally at up to 80 mg/kg, they protected the myocardium against ischemic insult through increasing enzymatic and nonenzymatic antioxidant (SOD, CAT, GPx, T-AOC) levels in vivo, and by decreasing oxidative stress damage. The intra-peritoneal administration of a steroidal saponin isolated from the rhizomes of *D. nipponica*, trillin, in rats exerted beneficial effects in improving the levels of lipid peroxidation and SOD activity [28].

Various other health benefits of yam preparations have also been reported. Total steroid saponins (TSS) of a *D. zingiberensis* rhizome containing 36 quantified compounds [161] alleviated arthritic progression in adjuvant-induced arthritis (AIA) rats in vivo, which correlated with suppression of the overproduction of inflammatory cytokines, oxidant stress makers, eicosanoids, and inflammatory enzymes, while the release of SOD and IL-10 increased [19]. Zhou et al. [145] reported that monosodium-urate-crystals-induced gouty arthritis in rats may be treated with Chinese herbal medicine (*D. nipponica*) by regulating lysosomal enzymes, antioxidant capacities and the NALP3 inflammasome; the authors demonstrated that the total saponin groups could increase the activities of GPx and SOD, and reduce the activities of several enzymes and the content of several important inflammatory biomarkers. Orally administered TS from *D. nipponica* remarkably decreased MDA, iNOS and ^•^NO levels in mice, while the levels of GSH, GSH-Px and SOD increased [162]. Water extract from *D. opposita* significantly increased plasma SOD activity and decreased plasma MDA concentration in hypertensive rats [163]. EtOH (95%) extract from Taiwanese yam (*D. japonica*, var. *pseudojaponica*) tubers decreased the LPS-induced ^•^NO production and expressions of iNOS and COX-2 in RAW264.7 cells; moreover, they had effects on the activities of CAT, SOD, and GPx, as well as on the levels of MDA in the edematous-paw mice [142]. The TPC, TFC, and radical-scavenging capacities of the extract were determined (Table 2 and Table 3). In the livers of mice treated with the EA fraction of *D. bulbifera* rhizome, the level of LPO increased remarkably, while the SOD, GPx, GST, GR, and glutamate-cysteine ligase (GCL) of the hepatic tissues all decreased conspicuously [164]. Yam (*D. pseudojaponica*) was also found to increase the activities of SOD and GPx, and to decrease the MDA level in the brains of D-gal-treated mice [165]. Rats fed with a high-cholesterol diet supplemented with either 0.1% or 0.5% diosgenin, for 6 weeks, exhibited a decrease in TBARS, while SOD in the plasma and liver, GSH-Px in the erythrocytes, and CAT in the erythrocytes and liver were significantly increased in the 0.5% diosgenin group [166]. The activity of SOD and CAT was significantly higher in a probucol and Dioscorea rhizome group of rabbits than in the control group fed chow containing 0.5% cholesterol and 10% corn oil [167]. The pre-intake of Chinese (*D. alata*) and Japanese (*D. japonica*) yam containing polyphenols and flavonoids significantly alleviated subsequent LPS-induced oxidative injury in mice by decreasing lipid oxidation levels and fibronectin production, and elevating SOD activity [100]. Strong antioxidant (+)-catechin, which was present in the extract of *D. bulbifera* bulbils, was suggested as a major bioactive constituent responsible for wound healing by promoting the cell proliferation and cell migration of human fibroblasts [62]. Yam (*D. oppositifolia*) extract demonstrated antioxidant and protective effects on the liver of rainbow trout [168], and an extract of Chinese yam (*D. opposita*) activated the antioxidant defense system in oxidatively damaged TM3 cells and KDS-Yang rats [147].

## 4. Phytochemicals of *Dioscorea*

Phytochemicals were studied in various *Dioscorea* spp., with saponins, phenanthrenes, and anthocyanins being the most widely investigated compounds. Therefore, this section does not discuss the very widely investigated steroidal saponins diosgenin and dioscin: several recently published reviews are available on the health benefits of these compounds (204–210). This section includes only short comments on the most recently-published data, which are summarized in Table 5. The structures of some secondary metabolites important for *Dioscorea* are given in Figure 1.

Flavonoids and phenolic acids are important and are the most widely studied antioxidants in many plants. The analysis of anthocyanins using LCMS-IT-TOF mass spectrometry in the vines of purple yams (*D. alata*) revealed that two investigated accessions from 8 months old had cyanidin or peonidin nuclei, while their glycosides were nonacylated, monoacylated, or diacylated with sinapic or ferulic acid. Cyanidin 3-(6-sinapoyl gentiobioside (alatanin C) was the major yam anthocyanin [93]. The compounds identified in the colored tubers of purple sachapapa (*D. trifida*) were derived from peonidin (Pn), cyanidin (Cy), and pelargonidin (Pg) aglycones, most of them showing the same substitution pattern, anthocyanin 3-acylglucoside-5-glucoside, where the acyl residue consisted of a hydroxycinnamic acid [113].

Some studies applied more modern extraction methods for *Dioscorea*. Among the 17 phenolic compounds—quantified in *D. glabra* leaf extracts isolated using microwave-assisted extraction—rosmarinic acid was the major one (22.31 ± 1.33 mg/g edw), while rutin was the dominant flavonoid, followed by quercetin; the latter compound was found in the highest quantity (8.66 ± 0.29 mg/g DW) in *D. alata* leaves [152]. Ochoa et al. [92] optimized ultrasound-assisted extraction (UAE) conditions for ethanolic anthocyanin-rich extracts from purple yam (*D. alata*); the best results were obtained at 60 °C for 10 min with 60% amplitude and an EtOH-to-water ratio of 80:20. Using SuperPro Designer^®^ software (v. 8.5) (Intelligen Inc., Scotch Plains, NJ, USA) the authors determined that the cost of manufacturing decreased from 950.52 to 124.08 USD/kg of extract when the extractor capacity increased from 5 to 500 L; they concluded that UAE would be economically feasible when the selling price is above 170 USD/kg. It is evident that there is a need to expand the studies on using advanced techniques such supercritical and subcritical fluid, pressurized liquid extraction, enzyme assisted, and other methods for the recovery of bioactive constituents. The preservation of heat-sensitive antioxidants, meeting green chemistry principles, and sustainability aspects should be taken into account.

Phenolic acids and flavonoids (in total, 21 compounds) were quantified in 80% ethanol extract isolated from *D. alata* tubers at ambient temperature. Kaempferol (9.219 mg/100 g pdw) and myricetin (4.613 mg/100 g pdw) were major quantitative constituents, while the content of other identified phenolics was less than 1 mg/100 g pdw [94]. Anthocyanins are the main pigments in the purple yams, while according to the recent report of Zhao et al. [69], the yellow pigments of fresh-cut yam (*D. opposita*) slices were composed primarily of bisdemethoxycurcumin (73.7%) and two other unknown compounds.

Chaniad et al. [62] partitioned crude EtOH extract of *D. bulbifera* bulbils with HX, CF, EA, and water, and separated these fractions using chromatography; this resulted in the isolation of 8-epidiosbulbin E acetate; 15,16-epoxy-6α-*O*-acetyl-8β-hydroxy-19-nor-clero-13; 14-diene-17,12;18,2-diolide; sitosterol-β-*D*-glucoside; 3,5-dimethoxyquercetin; (+)-catechin; quercetin; kaempferol; allantoin; 2,4,3′,5′-tetrahydroxybibenzy; 2,4,6,7-tetrahydroxy-9,10 dihydrophenanthrene; and myricetin. Some of these compounds inhibited ^•^NO, DPPH^•^, and ^•^OH.

Boudjada et al. [31] isolated 7 phenanthrene and dihydrophenanthrene derivatives from *D. communis*: some of them inhibited acetylcholinesterase (IC_50_ = 69.41 μg/mL) and butyrylcholinesterase (IC_50_ = 11.40, 14.60 and 14.34 μg/mL). Paul et al. [21] reported sex-specific variations in phytochemicals and the antimicrobial potentiality of *D. alata*, *D. hamiltonii*, *D. oppositifolia*, *D. pubera*, and *D. wallichii* tubers. Some studies reported phytochemicals in different yam anatomical parts. Yu et al. [171] compared phytochemicals in the Chinese yam (*D. polystachya*) tuber flesh and tuber cortex. Among the 38 detected compounds, dehydroepiandrosterone was more abundant in the flesh, while allantoin and flavonoids were more abundant in the cortex. In addition, the authors determined that the levels of allantoin, adenosine, and glutamine increased with the growing years. 2,7-Dihydroxy-4,6-dimethoxy phenanthrene, 6,7-dihydroxy-2,4-dimethoxy phenanthrene and 6-hydroxy-2,4,7-trimethoxyphenanthrene (batatasin I) were identified in the EtOH extracts of yam tuber (*D. batatas*) peel, while in the flesh, they were not detected; these phenanthrene derivatives scavenged DPPH^•^ and ABTS^•+^ [54]. Yang et al. [80] isolated and identified 19 phytochemicals (dihydrostilbenes, phenanthrenes, diarylheptanoids, and apigenin) in the CF-soluble fraction of *D. opposita*, and some of them scavenged DPPH^•^ and ^•^O_2_^−^.

Itharat et al. [175] isolated eight compounds from an EtOH extract of *D. membranacea*, two naphthofuranoxepins (dioscorealides A and B); a 1,4-phenanthraquinone (dioscoreanone); three steroids (β-sitosterol, stigmasterol and β-*D*-sitosterol glucoside); and two steroid saponins, diosgenin-(3-*O*-α-L-rhamnopyranosyl (1→2)-β-*D*-glucopyranoside and diosgenin 3-*O*-β-*D*-glucopyranosyl (1→3)-β-*D*-glucopyranoside). Their DPPH^•^ scavenging capacity at 0.1 mg/mL ranged from 0.8 to 33.1%. Nine compounds were isolated from the 75% ethanol extract of *D. septemloba* rhizomes, including new phenanthropyran, dioscorone B; moreover, a new phenanthrene named 2,20,6,60—tetramethoxy-4,40,7,70-tetrahydroxy-1,10-biphenanthrene showed excellent DPPH^•^-scavenging activities, with IC_50_ values of 0.07 ± 0.10 μM and 0.13 ± 0.09 μM, respectively [172]. Using UPLC-DAD-Q-TOF-MS/MS, Zhao et al. [29] detected 42 and 38 compounds in methanol extracts of *D. oppositifolia* (Chinese yam) and *D. hamiltonii*, respectively, while 14 of them were quantified using HPLC. In vitro propagation of *D. birmanica* shoots resulted in 0.37 ± 0.03% (*w*/*w*) of diosgenin-3-*O*-a-l-rhamnopyranosyl (1→2)–β-d-glucopyranoside, while its content in mother rhizomes was 3.27 ± 0.04% [108]. The β-Carotene content of MPYF (prepared using a modified process, boiled for polyphenol oxidase inhibition before drying)-substituted pasta was significantly higher compared to TPYF (prepared using a traditional process) samples [11]. One new bibenzyl and one new diarylheptanone and diobulbinone A, together with sixteen known compounds, were isolated form the rhizomes of *D. bulbifera*; this was accomplished by suspending it in H_2_O, successively extracting with 80% EtOH, petroleum ether, and ethyl acetate, and further fractionating with silica gel, Sephadex LH-20, and ODS [173]. It was reported that *Dioscorea* saponin (DS) can increase the exopolysaccharide production of tuber melanosporum [176].

Only a few reports are available on the analysis of *Dioscorea* extracts using gas chromatography including volatile constituents. 2-Acetyl furan and 2-acetylpyrrole had the highest odor activity (FD ≥ 128) factors in the roasted white yam (*D. rotundata*), while other compounds among the 29 aroma-active constituents with the FD factors ≥ 32 were 2-methylpyrazine, ethyl furfural, and 5-hydroxy methyl furfural [177]. Dey at al. [133] reported mainly higher fatty acids and their esters in an MeOH extract of freeze-dried tubers of *D. alata*’s various secondary metabolites. Alsawalha et al. [104] performed GC–MS analysis of an MeOH extract of *D. villosa* leaf powder; however, the identification results of the detected compounds, based on the raw MS software report, are too preliminary.

## 5. Conclusions

Numerous studies reported antioxidant properties in extracts and fractions isolated from different anatomical parts of *Dioscorea* spp. The majority of the published results were obtained in studies on plant tubers/rhizomes, while reports on leaves are rather scarce. Some studies focused on plant bulbils as well. Antioxidant potential studies applied mainly to in vitro chemical and bioassays using cell cultures, while in vivo studies with animal models were also available, albeit in smaller numbers. Some of the reported studies supported radical-scavenging capacity and other antioxidant activities of *Dioscorea* by investigating the phytochemical composition of the extracts and fractions, isolated using various solvents and methods. In general, leaf extracts were stronger antioxidants than rhizome/root preparations. Moreover, several studies demonstrated that tuber peel accumulate more antioxidants than its flesh. The antioxidant potential of different anatomical parts of *Dioscorea* may be related to their phytochemical composition. Steroidal saponins, phenanthrenes and anthocyanins (in purple yams) are important constituents in tubers, while phenolic acids and flavonoids are the main antioxidants determined in the leaves. Several studies also reported radical-scavenging properties in the mucilage, purified oligosaccharides and protein hydrolysis products obtained from rhizomes. Several in vivo studies using various animal models demonstrated the anti-inflammatory effects of *Dioscorea* preparations, which may be related to the increased levels of endogenous cell antioxidants, namely SOD, CAT, GPx, and reduced lipid peroxidation.

The reported data may assist in explaining ethnopharmacological knowledge regarding the health benefits of *Dioscorea* preparations. Moreover, such knowledge may foster a wider application of *Dioscorea* plants in developing new functional foods and nutraceuticals. In addition, it may be noted that, until now, published articles have focused mainly on the several most widely used species. A large number of *Dioscorea* spp. remain poorly investigated, and present a large pool for further studies.

## Figures and Tables

**Figure 1 molecules-27-02530-f001:**
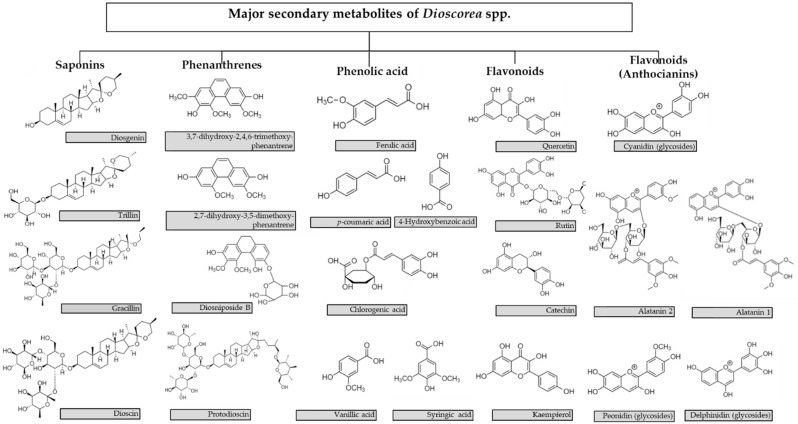
The structures of the main important *Dioscorea* secondary metabolites.

**Table 1 molecules-27-02530-t001:** Search results for available publications on *Dioscorea* (accessed on 21 December 2021).

Search Words	WoS		Science Direct		PubMed	
	Total	Review	Total	Review	Total	Review
*Dioscorea* (topic)	4779	218	5171	553	1665	61
*Dioscorea* (title)	2152	24	508	2	937	16
*Dioscorea* + antioxidant (topic + topic)	509	26	257	41	203	12
*Dioscorea* + antioxidant (title + topic)	237	7	109	2	108	1
*Dioscorea* + antioxidant (title + title)	73	1	na *	na	0	0

* na, not applicable.

**Table 2 molecules-27-02530-t002:** Antioxidant characteristics of various *Dioscorea* spp.

Species: Plant Part	Solvent, Analyzed Product	Antioxidant Characteristics	Ref.
*D. alata* (2)/*D. bulbifera/D. batatas/D. nipponica*: rhizome	MeOH-E	DPPH^•^ (IC_50_, µg/mL): 142.30 ± 2.58 and 486.43 ± 8.45/421.70 ± 17.24/432.66 ± 8.07 and 403.16 ± 14.59/371.64 ± 12.59 (AA, 10.77 ± 0.13; α-Toc, 40.24 ± 6.98; BHT, 11.92 ± 10.67)	[45]
*D. alata*: dried tubers	EtOH (80%)-E	DPPH^•^ (IC_50_, mg/gm dw): 0.603 ± 0.010 ABTS^•+^ (IC_50_, mg/gm dw): 0.136 ± 0.001	[46]
*D. alata*: flour/paste	H_2_O 1:10 (*w/v*)	DPPH^•^ (IC_50_, mg/mL): 18.52 ± 2.1/17.86 ± 0.9 ABTS^•+^ (mmol TE/100 g): 1.04 ± 0.00/1.15 ± 0.01 Fe^3+^ RP (mmol AAE/100 g): 0.96 ± 0.06/1.18 ± 0.10 ^•^OH (%): 49.37 ± 1.52/53.88 ± 4.59 Fe^2+^ chelating (%): 11.13 ± 2.77/18.10 ± 1.38	[47]
*D. alata*: purple yam	Effect of processing	DPPH^•^ (%): 79.08 (raw), 61.75 (blanched), 40.75 (washed), 32.16 (dried), 30.01 (flour)	[48]
*D. alata*: flour of tubers	H_2_O + AcA (5%)-E	DPPH^•^ (EC_50_, mg/mL): 2.55 to 8.70, depending on origin and treatment	[49]
*D. alata*: bulb	MeOH-E	DPPH^•^ (IC_50_, μg/mL): 14.68 (AA 24.95) Fe^3+^ RP (max absorption at 1 mg/mL): 1.317; BHT 1.472	[50]
*D. alata* (2 var); *D. doryophora*; commercial	Yam flour	ORAC (μmol TE/g): *Tai-nung*, 119.34; *Ta-shan*, 65.42 *Hang-chun* 60.63 103.17	[51]
*D. alata*: dried tubers of Chinese purple yam	EtOH (80%)-E: flesh/peel	DPPH^•^ (IC_50_, µg de/mL): 183.4 ± 5.3/47.7 ± 2.7; FRAP (mg FeSO_4_·7H_2_O/g de): 86.5 ± 1.6/144.5 ± 8.5 ABTS^•+^ (mg TE/g de): 108.1 ± 2.8/357.7 ± 8.4 ^•^O_2_^−^ (IC_50_, μg de/mL): 457.5 ± 20.8/162.2 ± 9.8 ChelA (%): 24.3 ± 1.6/21.4 ± 0.2	[52]
	EA fr: flesh/peel	DPPH^•^: 13.9 ± 1.1/8.5 ± 0.2; FRAP: 630.1 ± 19.4/534.4 ± 35.3 ABTS^•+^: 1326.1 ± 17.3/1578.1 ± 15.5; ^•^O_2_^−^: 78.2 ± 1.2/30.3 ± 0.3 ChelA: 15.1 ± 1.1/8.5 ± 1.0	
	BuOH fr: flesh/peel	DPPH^•^: 245.1 ± 32.1/43.8 ± 3.3; FRAP: 83.9 ± 5.3/162.1 ± 3.0; ABTS^•+^: 111.8 ± 5.5/448.4 ± 9.0; ^•^O_2_^−^: 333.2 ± 10.6/147.9 ± 4.1 ChelA: 0.0 ± 0.1/34.3 ± 3.2	
	HX fr: flesh/peel	DPPH^•^: 632.0 ± 26.9/55.0 ± 0.9; FRAP: 42.3 ± 1.2/97.3 ± 11.2; ABTS^•+^: 39.8 ± 1.6/242.3 ± 14.4; ^•^O_2_^−^: 1387.5 ± 110.3/237.9 ± 12.2; ChelA: 32.0 ± 2.2/34.1 ± 1.2	
	Remaining H_2_O fr: flesh/peel	DPPH^•^: 1532.7 ± 123/225.1 ± 12.9; FRAP: 8.5 ± 0.5/29.1 ± 1.7; ABTS^•+^: 12.2 ± 0.3/74.0 ± 2.9; ^•^O_2_^−^: 874.5 ± 18.2/461.1 ± 14.1 ChelA: 50.4 ± 5.1/29.1 ± 0.6	
*D. alata*: tubers (3 cultivars from Taiwan)	Hot pressurized EtOH-E: flesh/peel	DPPH^•^ (EC_50_, µg/mL): ND/86.6–305.4	[53]
	fr of HX/CF/EA/BuOH/H_2_O	Flesh: 328–2360/144.6–549.4/112.8–343.6/288.1–1526/ND Peel: 133.0–932.7/45.8–136.6/14.5–38.8/67.2–678.3/ND	
*D. batatas*: tubers, 3 purified phenanthrenes	EtOH (95%)-E flesh/peel	DPPH^•^ (IC_50_, mg/mL): 7.68/0.944 ABTS^•+^ (IC_50_, mg/mL): 3.43/0.771	[54]
	2,7-dOH-4,6-dMetOP/6,7-dOH -2,4-dMetOP/6-OH-2,4,7-tMetOP	DPPH^•^ (IC_50_, mg/mL): 0.0645/0.154/0.566 ABTS^•+^ (IC_50_, mg/mL): 0.0482/0.153/0.297	
*D. batatas*: dried and smashed into raw yam meals	EtOH (70%)/MeOH (70%)/CF: MeOH (2:1)	DPPH^•^ (IC_50_, mg/mL): 1.22 ± 0.03/1.34 ± 0.02/0.71 ± 0.00 ABTS^•+^ (IC_50_, mg/mL): 2.08 ± 0.20/2.24 ± 0.10/1.15 ± 0.05 FRAP (μM Fe^2+^ at 1.0 mg/mL): 108.3 ± 0.0/91.67 ± 0.48/220.6 ± 0.7	[55]
	EtOH (70%)/MeOH (70%)/CF: MeOH (2:1)	Fe^2+^ chelating: 0.12 ± 0.02/0.09 ± 0.01/0.97 ± 0.03 NO^•^: 0.46 ± 0.02/0.45 ± 0.00/0.55 ± 0.02 β-carotene bleaching: 0.15 ± 0.04/0.07 ± 0.01/0.14 ± 0.01 LPI: 0.50 ± 0.01/0.58 ± 0.00/0.05 ± 0.01(IC_50_, mg/mL for all)	[56]
*D. batatas*: raw yam	BuOH/EA	ABTS^•+^ (IC_50_, mg/mL): 0.70 ± 0.01/0.45 ± 0.01 DPPH^•^ (IC_50_, mg/mL): 0.50 ± 0.00/0.34 ± 0.01; FRAP (μM Fe^2+^) ~347/~560 at 1 mg/mL	[57]
*D. batatas*: raw yam	BuOH/EA	Fe^2+^ chelating: 0.64 ± 0.01/0.70 ± 0.01 NO^•^: 0.53 ± 0.01/0.26 ± 0.03; NO_2_ scavenging: 1.92 ± 0.03/3.92 ± 1.00 β-carotene bleaching: 0.08 ± 0.01/0.05 ± 0.01 LPI: 0.02 ± 0.01/0.01 ± 0.00 (IC_50_, mg/mL for all assays)	[58]
*D. batatas*: thermally treated yam	BuOH/EA	ABTS^•+^ (IC_50_, mg/mL) 1.31 ± 0.05/0.98 ± 0.02 DPPH^•^ (IC_50_, mg/mL) 0.70 ± 0.00/0.74 ± 0.01 FRAP (μM Fe^2+^): 237.86 ± 4.07/244.05 ± 9.37	[59]
*D. batatas*: tubers (thermally treated yam)	BuOH/EA	Fe^2+^ chelating: 0.11 ± 0.02/0.81 ± 0.01 ^•^NO: 0.73 ± 0.01/0.45 ± 0.02 NO_2_ scavenging: 3.78 ± 1.24/3.87 ± 0.40 β-carotene bleaching: 0.11 ± 0.01/0.11 ± 0.00 LPI: 0.02 ± 0.01/0.05 ± 0.01 (IC_50_, mg/mL for all assays)	[60]
*D. birmanica*: freeze-dried rhizomes	EtOH (95%) extract	(EC_50_, μg/mL) DPPH^•^, 8.53 ± 1.32; ABTS^•+^, 21.56 ± 1.72; ^•^O_2_^−^, 50.91 ± 0.39; ^•^NO, 26.93 ± 4.79; LPI: 33.37 ± 2.88. FRAP (mg Fe^2+^ eq/g extract): 406.96 ± 11.33 Fe^2+^ chelating (EC_50_, mg/mL): 1.06 ± 0.03	[16]
*D. bulbifera*: tuber	MeOH (80%)	DPPH^•^ (IC_50_, μg/mL): 261.09; AA, 10.65; ^•^O_2_^−^, 2089.3; Q, 17.01	[61]
*D. bulbifera*: bulbils	EtOH/H_2_O crude extracts	DPPH^•^ (IC_50_, μM): 34.14 ± 0.68/13.20 ± 0.77 ^•^OH (IC_50_, μM): 79.00 ± 0.78/>100	[62]
	CF_/_EA/H_2_O fr of EtOH	DPPH^•^: 13.35 ± 0.37/14.00 ± 0.36/18.83 ± 0.46 ^•^OH: >100/37.04 ± 0.50/39.31 ± 0.42	
*D. bulbifera*: leaves	MeOH/EA/HX extracts	DPPH^•^ (%): 79.0 ± 0.31/23.2 ± 0.05/11.5 ± 0.31 ABTS^•+^ (mg AAE/g): 65.6 ± 0.35/59.5 ± 0.10/14.9 ± 0.05 FRAP (mM/Fe^2+^g dw): 31.34 ± 2.06/10.98 ± 0.64/9.50 ± 0.48	[63]
*D. bulbifera*: bulb	PE/EA/MeOH (sequentially)/EtOH (70%)	DPPH^•^ (%): 61.82 ± 1.55/82.79 ± 1.24/76.11 ± 1.26/80.64 ± 1.24 ^•^O^−^_2_ (%): 26.88 ± 1.28/57.60 ± 0.81/59.75 ± 0.98/54.76 ± 1.20 O_2_^−^ (%): 28.30 ± 0.36/59.24 ± 1.44/59.65 ± 1.41/57.34 ± 1.41 ^•^NO (%): 20.57 ± 0.57/54.55 ± 0.21/57.59 ± 0.64/49.85 ± 0.16 ^•^OH (%): 44.51 ± 0.49/66.67 ± 0.73/76.11 ± 1.26/64.23 ± 1.25	[64]
*D. bulbifera*: tubers	Pre-purified/crude polysaccharide	DPPH^•^ (mg IE/g): 0.28 ± 0.01/0.94 ± 0.02 ABTS^•+^ (mg TE/g): 623.33 ± 4.71/165.00 ± 2.36 FRAP (mM Fe^2+^/g): 0.175 ± 0.001/1.056 ± 0.001	[65]
*D. bulbifera*: tubers	Pre-soaking in 0–10% oligosaccharide	DPPH^•^ (0.1 mL): ~17–27% ABTS^•+^ (0.1 mL): ~23–45% (determined from figure)	[66]
*D. hirtiflora*: tubers	Successively DCM/EA/MeOH	DPPH^•^ (IC_50_, μg/mL): 49.7 ± 0.97/11.9 ± 0.85/11.8 ± 0.23	[67]
*D. dumetorum*		89.0 ± 5.10/103.2 ± 6.9/137 ± 5.90	
*D. bulbifera* (mauve)		46.7 ± 1.57/14.6 ± 0.90/29.9 ± 0.68	
*D. bulbifera* (yellow)		57.7 ± 1.32/64.1 ± 0.89/68.6 ± 8.50 (AA, 6.90; GA, 8.60)	
*D. caucasica*: freeze-dried leaves	EtOH (70%)	DPPH^•^ (mg TE/g edw): 279 ± 4 ABTS^•+^ (mg TE/g edw): 880 ± 10	[68]
*D. communis*: rhizome	DE/EA crude extracts	DPPH^•^: 8.7 ± 0.9/40.1 ± 0.2; ABTS^•+^, 7.6 ± 0.0/11.6 ± 0.0; FRAP, 44.8 ± 3.6/79.0 ± 0.0; CUPRAC, 10.01 ± 0.2/20.8 ± 0.5 (all IC_50_, µg/mL)	[31]
	3 purified phenanthrenes	DPPH^•^, <200/61.2 ± 1.1/6.0 ± 0.2; ABTS^•+^, <200/19.60 ± 0.0/2.4 ± 0.1; FRAP, <50/<50/9.9 ± 1.0; CUPRAC, <200/194.0 ± 0/15.0 ± 0.5 (all IC_50_, µg/mL)	
*D. opposita*: fresh-cut yam	H_2_O/MeOH/EA-E	ORAC (µmol TE/g): ~33/63/48 (white); ~29/96/73 (yellow) DPPH^•^ (mmol TE/g): ~11/10/8.5 (white); ~9/15/8.5 (yellow) (determined from the figures)	[69]
*D. opposita*: purified yellow pigment	MeOH-Amberlite XAD-7-Sep-Pak C18	^•^OH scavenging (IC_50_ mg/mL): 0.098 ± 0.032 (AA, 1.21 ± 0.0)	[70]
*D. hamiltonii*: herb	MeOH-E	Fe^3+^ RP (mg AAE/g): 3.30 at 0.5 g/mL	[29]
*D. opposita*: herb	MeOH-E	Fe^3+^ RP (mg AAE/g): 4.71 at 0.5 g/mL	[29]
*D. opposita*: herbal medicine product	EtOH (70%)-E 5 g/50 mL	DPPH^•^ (%, 0.5 mL): 43.2 ± 2.35; ABTS^•+^ (%, 20 μL): 40.01 ± 3.0; SOD (%, 0.2 mL): 39.97 ± 8.87	[71]
*D. hemsleyi*: rhizome	Cold/warm/hot H_2_O extracted polysaccharides	DPPH^•^ (IC_50_, mg/mL): 4.56 ± 0.15/6.95 ± 0.13/8.85 ± 0.16 Fe^3+^ RP (mg AAE/g): 42.98 ± 0.79/31.78 ± 0.35/25.64 ± 0.24 FRAP (mg AAE/g): 13.88 ± 0.54/8.91 ± 0.18/5.70 ± 0.05	[72]
*D. nipponica*: rhizome	MeOH-E	DPPH^•^ (µg/mL): 371.64 ± 12.30	[45]
*D. nipponica*: rhizome	H_2_O-soluble polysaccharide	^•^OH scavenging (%): 3.35–43.73 at 0.25–4 mg ^•^O_2_^−^ (%): 27.5–35.52 at 0.25 mg and 2 mg	[73,74]
*D. nipponica*: leaves	EtOH (70%)-E	(mg TE/g edw) DPPH^•^: 415 ± 9; ABTS^•+^: 659 ± 4	[68]
*D. opposita*: rhizome, flesh/peel	Hot H_2_O-E	DPPH^•^ (IC_50_, μg/mL): 1008.62 ± 5.96/374.85 ± 6.78 ^•^OH (IC_50_, μg/mL): 1267.04 ± 5.13/744.25 ± 3.46	[75]
	EtOH (80%)-E	DPPH^•^ (IC_50_, μg/mL): 897.14 ± 4.73/415.74 ± 3.79 ^•^OH (IC_50_, μg/mL): 1155.00 ± 9.64/845.21 ± 14.66	
*D pentaphylla*: leaves	EA (80%)/MeOH	DPPH^•^ (IC_50_, μg/mL): 135.12 ± 0.95/85.61 ± 0.64 ^•^NO (IC_50_, µg/mL): 195.78 ± 0.29/68.13 ± 0.26	[76]
*D pentaphylla*: tubers	MeOH/Ac crude extracts	DPPH^•^ (EC_50_, μg/mL): 82.07 ± 0.08/89.41 ± 0.39 Metal chelating (EC_50_, μg/mL): 81.47 ± 0.36/86.52 ± 0.55	[77]
*D. japonica*: tubers	EtOH (70%)	DPPH^•^: at 0.1–10 mg/mL from 18.01 to 89.64%	[78]
*D. villosa*: leaves	H_2_O/MeOH-E	DPPH^•^ (IC_50_ μg edw/mL): 21.36/40.24	[79]
*D. rotundata*: flour/paste	H_2_O 1:10 (*w/v*)	DPPH^•^ (IC_50_, mg/mL): 19.26 ± 2.4/19.56 ± 1.5 ABTS^•+^ (mmol TE/100 g): 0.79 ± 0.00/1.31 ± 0.08 Fe^3+^ RP (mmol AAE/100 g): 0.93 ± 0.58/1.45 ± 0.47 ^•^OH (%): 56.71 ± 1.51/63.72 ± 2.31 Fe^2+^ chelating (%): 8.33 ± 2.78/11.11 ± 5.56	[47]
*D. opposita*: rhizomes	18 compounds from MeOH-E	DPPH^•^ (EC_50_, µg/mL): 12.3 ± 0.2->100 (AA = 19.2) ^•^O_2_^−^ (EC_50_, µg/mL): 38.8 ± 1.3->100 (AA = 16.7)	[80]
*D. opposita*: tuber mucilage	H_2_O-E	(%) DPPH^•^ 38.2 ± 3.14/^•^O_2_^−^, 84.1 ± 6.57/^•^OH, 79.4 ± 6.42; 1 mM AA: 94.7 ± 3.21/89.9 ± 5.31/16.1 ± 0.64	[81]
*D. schimperiana*	MeOH (60%)	ABTS^•+^ (MTE/100 gMF): 0.153 (yellow); 0.218 (with red dot); 0.151 (red fleshed)	[82]
*D. bulbifera*	EtOH (70%)	DPPH^•^ (%): 64.81 ± 2.80; ABTS (%): 72.44 ± 5.28	[83]
*D. polystachya*		DPPH^•^ (%): 77.09 ± 0.00; ABTS (%): 99.89 ± 1.60	
*D. batatas*		DPPH^•^ (%): 75.74 ± 0.94; ABTS (%): 83.66 ± 9.03	
*D. quinqueloba*		DPPH^•^ (%): 84.46 ± 0.41; ABTS (%): 95.56 ± 0.96	
*D. batatas*: thermally treated yam	MeOH (70%)/EtOH (70%)/CF: MeOH (2:1)	DPPH^•^ (IC_50_, mg/mL): 0.56/0.52/0.43 ABTS^•+^ (IC_50_, mg/mL): 1.28/1.05/0.60	[84]
*D. opposita*: tuber mucilage	Autolysis/hydrolysis (pepsin/trypsin/papain)	^•^O_2_^-^ (all in % at 100 mg/mL): 60.2 ± 4.01/82.2 ± 5.95 (p)/56.0 ± 4.36 (t)/98.5 ± 3.54 (p)/52.6 ± 4.18 ^•^OH: 90.4 ± 5.25/91.2 ± 5.86 (p)/91.2 ± 5.50 (t)/91.6 ± 5.92 (p)/67.6 ± 4.34 DPPH^•^: 75.2 ± 4.77/61.7 ± 4.03 (p)/87.1 ± 5.04 (t)/70.2 ± 4.89 (p)/87.6 ± 2.75 (all in mmol α-Toc)	[85]
*D. batatas*: yam	MeOH-E and fractions/HX/EA/BuOH/H_2_O	DPPH^•^ (IC_50_, µg/mL): 602.2 ± 71.92/510.6 ± 25.02/80.5 ± 12.37/263.0 ± 56.47/>1000	[86]
Arial bulbils	MeOH-E and fractions/HX/EA/BuOH/H_2_O	DPPH^•^ (IC_50_, µg/mL): 376.3 ± 32.18/180.9 ± 24.77/38.1 ± 5.82/161.4 ± 32.14/>1000 AA: 15.2 ± 2.96; BHT 18.6 ± 4.05; VitE 35.6 ± 5.12	[87]
*D. alata*: yam	EtOH (50%)/hot H_2_O/H_2_O	DPPH^•^ (mg α-Toc eq/g): 4.14 ± 0.01/3.71 ± 0.03/3.37 ± 0.04 (peel); 0.73 ± 0.07/0.72 ± 0.03/0.22 ± 0.03 (flesh) Fe^3+^ RP (mg GAE/g): 41.3 ± 0.8/28.7 ± 0.4/30.6 ± 0.4 (peel); 0.58 ± 0.00/0.83 ± 0.02/0.86 ± 0.01 (flesh)	[88]
*D. japonica*: bulbil	MeOH (80%)-E: CF/EA/BuOH/H_2_O Fr/AA	DPPH^•^ (µg): 200.8 ± 7.9: 38.8 ± 3.6/14.8 ± 0.6/75.4 ± 1.6/>1000/3.3 ABTS^•+^ (mg): 2.3 ± 0.2: 0.5 ± 0.02/0.13 ± 0.02/0.93 ± 0.1/9.4 ± 0.5/1.2 µg	[89]
*D. batatas*: bulbil		DPPH^•^ (µg): 84.0 ± 2.6: 23.6 ± 2.0/9.2 ± 0.2/27.6 ± 0.8/>1000/3.3 ABTS^•+^ (mg): 0.9 ± 0.04: 0.3 ± 0.04/0.09 ± 0.04/0.42 ± 0.04/7.6 ± 0.4/1.2 µg	
*D. trifida*: dried tubers		ABTS^•+^ (µmol TE/100 g): 131.14 ± 5.49 to 174.52 ± 0.78 (depending on drying method)	[90]
*D. trifida*: tubers		(IC_50_, mg/mL) DPPH^•^: 7.44; ABTS^•+^: 0.54; ^•^O_2_^−^: 13.67	[91]

Abbreviations: AA—ascorbic acid; AAE—ascorbic acid equivalent; Ac—acetone; ABTS^•+^—2,2′-azinobis-(3-ethylbenzothiazoline-6-sulfonate) scavenging capacity; AcA—acetic acid; AAE—ascorbic acid equivalent; BHT—butyl hydroxy toluene; BuOH—*n*-butanol; CF—chloroform; ChelA—chelating activity; CUPRAC—cupric reducing antioxidant capacity; DCM—dichloromethane; de—dried extract; DE—diethyl ether; DPPH^•^—(2,2-diphenyl-1-picrylhydrazyl) scavenging capacity; dw—dry weight; E—extract; EA—ethyl acetate; edw—extract dry weight; eq—equivalent; EtOH—ethanol; Fe^2+^,Fe^3^^+^—ferric ion; fr—fraction; FRAP—ferric reducing antioxidant power (determined using ferric-tripyridyltriazine complex); Fe^3+^ RP—ferric reducing power (determined using *potassium ferricyanide complex*); GA—gallic acid; GAE—gallic acid equivalents; H_2_O—water; HX—hexane; IC_50_—effective concentration reducing 50% of the radicals present in the reaction; LPI—lipid peroxidation inhibition; MeOH—methanol; ND—not detected; NO_2_—*nitrogen dioxide* radical; ^•^NO—nitric oxide scavenging activity; ^•^O_2_^−^—superoxide anion scavenging activity; O_2_^−^—superoxide radical-scavenging activity; ^•^OH—hydroxyl radical-scavenging activity; ORAC—oxygen radical absorbance capacity; pdw—plant dry weight; PE—petrol ether; Q—quercetin; SOD—superoxide dismutase; TE—Trolox (6-hydroxy-2,5,7,8-tetramethylchroman-2-carboxylic acid) equivalent; α-Toc—α-tocopherol; 2,7-dOH-4,6-dMetOP—2,7-dihydroxy-4,6-dimethoxy phenanthrene; 6,7-dOH-2,4-dMetOP—6,7-dihydroxy-2,4-dimethoxy phenanthrene; 6-OH-2,4,7-tMetOP—6-hydroxy-2,4,7-trimethoxyphenanthrene (See Appendix A abbreviations).

**Table 3 molecules-27-02530-t003:** Total content of various groups of compounds in *Dioscorea* spp.

Species: Plant Part	Solvents	Characteritics	Ref.
*D. alata*: tubers	MeOH	TPC: male 12.21 ± 0.82; female: 17.53 ± 1.30 TFC: male 14.80 ± 0.69; female 9.17 ± 0.3	[21]
	EtOH (20%) fr.	TS (%): male: 0.48 ± 0.06; female: 0.93 ± 0.17	
*D. alata*: soaked flour of tubers	H_2_O + AcA (5%): in H_2_O/NaHSO_3_ 0.2%)/AA (0.1%)	TPC: 11.6 ± 0.31/12.7 ± 0.67/11.5 ± 0.13 TFC: 7.0 ± 0.14/7.4 ± 0.21/7.3 ± 0.23 TAC (mg CGE/100 g): 46.7 ± 3.35/74.3 ± 3.83/50.0 ± 2.26	[49]
*D. alata*: tubers	H_2_O	TPC: 1.00–3.85; TFC: 0.60–1.60; TAC: 0.10–0.90	[93]
*D. alata*: yam	EtOH (50%)/hot H_2_O/H_2_O	TPC (peel): 11.14 ± 0.30/6.09 ± 0.08/6.60 ± 0.13 TPC (flesh): 0.25 ± 0.01/0.40 ± 0.02/0.24 ± 0.00	[88]
*D. alata*: tubers	EtOH (80%)	TPC (mg GAE/g edw): 63.85 ± 1.83; TFC (mg RE/g edw): 8.21 ± 0.02; TFL (mg QE/g edw): 17.23 ± 0.19	[94]
*D. alata*: purple yam	Effect of processing	TAC (mg/100 g): 38.12 (raw), 36.73 (blanched), 32.63 (washed), 29.29 (dried), 27.27 (flour)	[48]
*D. glabra*: tubers		TPC (mg/100 g): 335.64 ± 3.92; TFC: 65.73; C: 23.49 ± 0.0413; H_2_O-soluble B: 0.036 to 4.159	[46]
*D. alata*: tubers	MeOH (50%) + 0.1% HCl	TAC: 3.32 mg/g	[95]
*D. alata*: tubers	MeOH (80%)	TPC (mg GAE/g edw): flesh 48.3 ± 4.1; peel 194.8 ± 14.6	[52]
	EA fr	TPC (mg GAE/g edw): flesh 479.5 ± 33.1; peel 695.1 ± 35.1	
*D. alata*: tubers	MeOH	TPC (mg GAE/g edw): 222.99; TFC (mg QE/g edw): 98.95	[50]
*D. batatas*: tubers; thermally treated yam	EtOH (70%)/MeOH (70%)/ CHCl_3_:MeOH (2:1)	TPC (mg CE/g edw): 43.38 ± 0.66/37.62 ± 0.88/67.17 ± 0.12	[55]
*D. batatas*: thermally treated		TS (mg/g dw): 42.52 ± 1.88 TT (mg CE/g dw): 14.95 ± 0.98	[56]
*D. batatas*: raw yam	BuOH/EA	TPC (mg CE/g edw): 78.68 ± 0.58/111.88 ± 0.66	[57]
*D. batatas*: tubers thermally treated	BuOH/EA	TPC (mg GE/g): 53.83 ± 1.00/51.63 ± 2.63	[59]
*D. batatas*: aerial bulbils	MeOH-E: and fr HX/EA/BuOH/H_2_O	TPC (mg/g): 60.60: 12.63/27.48/17.18/4.03 TFC (mg/g): 16.4: 32.1/70.1/34.2/1.38	[87]
*D. batatas*: thermally treated yam	MeOH (70%)/EtOH (70%)/CF:MeOH (2:1)	TPC: 63.63 ± 0.33/69.47 ± 1.00/97.49 ± 0.66	[84]
*D. batatas*: rhizome	MeOH: Jang-Ma/Dang-Ma	TPC (mg/g): 34.86 ± 0.15/45.84 ± 0.34; TFC (mg/g): 6.67 ± 0.22/7.33 ± 0.14; Total sugar (mg/g): 281.96 ± 0.08/140.86 ± 0.21	[45]
*D. alata*: rhizome	MeOH: Dungkun-Daema/Jasak-Ma	TPC (mg/g): 87.05 ± 0.0.11/27.98 ± 0.25; TFC (mg/g): 12.67 ± 0.34/7.75 ± 0.23; Total sugar (mg/g): 184.98 ± 0.14/107.61 ± 0.32	[45]
*D. batatas*: yam	MeOH-E: and fr HX/EA/BuOH/H_2_O	TPC (mg/g): 5.05: 2.61/48.31/8.49/3.81 TFC (mg/g): 4.85: 4.85/42.55/1.71/0.66	[86]
*D. hispida*:	MeOH-E: fr PE/CTC/DCM/H_2_O	TPC: 160.65 ± 0.18: 280.09 ± 0.54/287.50 ± 0.71/68.98 ± 1.43/22.99 ± 0.54	[23]
*D. bulbifera*: stem tuber	MeOH (80%)	TPC (mg GAE/mg edw): 0.243 ± 0.052; nontannins, 0.632 ± 0.048; tannins, 0.259 ± 0.034. TFL (mg QE/mg edw) 1.399 ± 0.075; TFC, 0.060 ± 0.025	[61]
*D. bulbifera*: rhizome	MeOH: Buchae-Ma	TPC (mg/g): 51.11 ± 0.16; TFC (mg/g): 10.33 ± 0.09; total sugar (mg/g): 179.79 ± 0.14	[45]
*D. nipponica*: rhizome	MeOH: Dungkun-Ma	TPC (mg/g): 52.08 ± 0.24; TFC (mg/g): 13.99 ± 0.11; total sugar (mg/g): 147.67 ± 0.09	[45]
*D. birmanica*: rhizome	EtOH (95%).	TPC (mg GAE/g e): 170.85 ± 3.02 TFC (mg CE/g e): 132.55 ± 3.59	[16]
*D. birmanica*: rhizome	EtOH (95%).	TPC (mg GAE/g e): 170.85 ± 3.02 TFC (mg CE/g e): 132.55 ± 3.59	[16]
*D. bulbifera*: tubers	MeOH	TPC (mg/100 g FW): 67.17 ± 0.12	[96]
*D. bulbifera*: tubers	Soaked in 0–10% oligosaccharide solution	TPC (mg GAE/g): 1.37 ± 0.3–1.41 ± 0.1 TFC (mg RE/g): 0.97 ± 0.1–1.04 ± 0.1	[66]
*D. bulbifera*: bulb	PE/EA/MeOH (sequentially)/EtOH (70%)	TPC (mg/mL): 49.22 ± 0.80/98.00 ± 1.17/145.4 ± 3.29/85.89 ± 1.16 TFC (mg/mL): 4.95 ± 0.1/27.86 ± 0.18/12.76 ± 0.48/12.10 ± 0.05	[64]
*D. hamiltonii*: tuber	MeOH	TPC: male 41.40 ± 2.94; female: 50.70 ± 2.49	[21]
		TFC: male: 25.67 ± 0.93; female: 36.67 ± 0.99	
	EtOH (20%) and its fraction	TS (%): male 0.95 ± 0.14; female 1.16 ± 0.18	
*D. hamiltonii* (syn *D. persimilis*): herbs	MeOH	TPC (µg/mL GAE): 158.21; TFC (μg/mL CE): 72.3	[29]
	Vanillin-CH_3_COOH and HClO_4_ mixture 1:5 (*v/v*)	TS (dioscin equivalents): 257.8 μg/mL	
*D. hispida*	MeOH-E: fr PE/CTC/DCM/H_2_O	TPC: 160.65 ± 0.18: 280.09 ± 0.54/287.50 ± 0.71/68.98 ± 1.43/22.99 ± 0.54	[23]
*D. oppositifolia*: tubers	MeOH	TPC: male: 11.03 ± 0.60; female: 13.65 ± 0.36 TFC: male: 7.21 ± 0.99; female: 15.03 ± 1.08	[21]
	EtOH (20%) and its fraction	TS (%): male: 0.45 ± 0.09; female: 0.81 ± 0.15	
*D. opposita*: tuber mucilage	Autolysis/hydrolysis	TPC (mg/g powder): 6.4 ± 0.08/15.3 ± 1.60 (pepsin)/11.2 ± 1.34 (trypsin)/7.4 ± 0.09 (papain)	[85]
*D. pubera*: tubers	MeOH	TPC (mg GAE/gm dw): male: 31.76 ± 0.21; female: 21.83 ± 2.5	[21]
	EtOH (20%) and its fraction	TFC (mg CAE/gm): male: 19.68 ± 1; female: 22.17 ± 0.2 TS (%): male: 0.88 ± 0.23; female: 0.92 ± 0.17	
*D. oppositifolia* (syn *D. opposita*): herb	MeOH-E	TPC (µg/mL GAE): 297.03; TFC (μg/mL CE) 49.6	[29]
	Vanillin-CH_3_COOH and HClO_4_ mixture 1:5 (*v/v*)	TS (μg/mL dioscin equivalents): 475.5	
*D. opposita*: rhizome	Hot H_2_O	TPC: flesh 1.77 ± 0.67; peel:10.97 ± 0.21; TFC (mg rutin/g eq): flesh: 1.03 ± 0.15; peel 1.77 ± 0.07; TC (mg/g): flesh: 324.90 ± 0.82; peel:123.50 ± 0.80	[75]
	EtOH (80%)	Flesh: TPC 7.77 ± 0.10; TFC (mg RE/g extract): 1.20 ± 0.10; TC (mg/g extract) 23.63 ± 0.45	
		Peel: TPC: 15.40 ± 0.10; TFC (mg RE/g extract): 2.62 ± 0.15; TC (mg/g extract) 17.60 ± 0.20	
*D. pentaphylla*: leaves	MeOH	TPC (mgGAE/g): 213.89 ± 3.93; TFC (mg QE/g): 41.5 ± 2.12	[76]
	EA	TPC (mgGAE/g): 76.39 ± 3.54; TFC (mg QE/g): 147.5 ± 3.54	
*D. schimperiana*: tubers flour	EtOH (20%) and its fraction; pasta with 60% yam flour	TPC: traditional process, 2.86 ± 0.02; modified process, 5.04 ± 0.03	[11]
*D. schimperiana*:	MeOH (60%)	TPC (mg/100 g): 10 (yellow); 8 (with red dot); 8 (red fleshed)	[82]
*D. trifida*: tubers	Commonly used methods	TPC (mg GAE/100 g): 187.09–513.67 ± 9.49 Total carbohydrate (%): 81.75 ± 0.24 Total starch (%): 74.11 ± 0.55	[90]
	EtOH (95%):HCl 85:15 (*v/v*)	TAC (mg C-3-Glc/100 g): 159.11–281.10 ± 0.01	
*D. trifida*: tubers	Dry powders	TPC (mg GAE/100 g dw): 166.10 ± 1.52; TFC (mg QE/100 g dw) 27.63 ± 2.69; TT (mg GAL/100 g dw) 9.62 ± 0.084; TAC (mg C-3-Glc/100 g dw): 21.59 ± 1.47	[91]
*D. wallichii*: dried powder	MeOH (male/female)	TPC: 10.73 ± 0.25/9.73 ± 0.28 TFC (mg CE/gm): 20.6 ± 0.6/26.00 ± 2.14	[21]
*D. quinqueloba*	H_2_O/MeOH/EtOH/ EA	TPC (mg/g): 10.16/10.48/14.67/9.91 TFC (mg/g): 7.58/9.91/10.58/16.02	[97]
*D. rotundata*: flour/paste	H_2_O 1:10 (*w/v*)	TPC (mg GAE/g): 1.56 ± 0.04/1.34 ± 0.02 TFC (mg QE/g): 0.16 ± 0.01/0.08 ± 0.01	[47]
*D. alata*: flour/paste		TPC (mg GAE/g): 1.38 ± 0.03/1.12 ± 0.02 TFC (mg QE/g): 0.18 ± 0.01/0.10 ± 0.02	
*D. japonica*: tubers	EtOH (70%)	TPC: 35.15 mg GAE/100 g edw	[78]
*Dioscorea* spp.: tubers, 5 cultivars	Inoculated with 6 spp. of arbuscular mycorrhizal fungi; flesh/peel	TPC (mg/kg): Tainung 1: 21.3-44.5/39.0-52.7; Tainung 2: 19.8–38.3/43.9–54.5; Ercih: 11.1–16.9/40.4–50.4	[98]
		TFC (mg/kg): Tainung 1: 4.1–8.6/9.9–15.1; Tainung 2: 4.4–6.9/9.6-10.4; Ercih: 4.5–5.6/7.8–10.9; Zigyuxieshu: 4.9–6.5/9.3–11.8 Tainung 5: 4.4–5.7/8.3–10.9	
		TAC (flesh/peel, mg/kg): Tainung 1, 2, Ercih: nd; Zigyuxieshu: 0.83–1.08/1.93–2.54; Tainung 5: 0.33–0.76/1.52–2.42	
*D. bulbifera*	EtOH (70%)	TPC: 2.23 ± 0.03; TFC (mg RE/g): 1.99 ± 0.17	[83]
*D. polystachya*		TPC: 3.65 ± 0.11; TFC (mg RE/g): 2.62 ± 0.20	
*D. batatas*		TPC: 2.25 ± 0.19; TFC (mg RE/g): 1.57 ± 0.06	
*D. quinqueloba*		TPC: 9.50 ± 0.38; TFC (mg RE/g): 1.30 ± 0.16	
*D. batatas*: raw	BuOH/EA	TPC (mg CAE/g-E): 78.68 ± 0.58/111.88 ± 0.66	[57]
*D. alata*: flour of 5 cultivars	Small/medium/large particle size fractions	Phenols (%): 0.27–1.39/0.52–2.82/0.48–2.20 TAC (mg/100 g): nd-14.20/nd-15.27/2.25–13.07 Carotenoids (µg/100 g): nd-132.12/nd-129.8/nd-123.1	[99]
*D. alata*: tubers		TPC (mg GAE/100 g): 157.7 ± 7.5; TFC (mg CE/100 g): 190.4 ± 10.9	[100]
*D. japonica*: tubers		TPC (mg GAE/100 g): 206.4 ± 12.8; TFC (mg CE/100 g): 178.2 ± 8.3	[100]
*D. bulbifera*: tuber		TPC (mg GAE/100 g FW): 166 ± 10	[96]
*D.versicolor*: tuber		TPC (mg GAE/100 g FW): 41 ± 5	
*D. deltoidea*: tuber		TPC (mg GAE/100 g FW): 15 ± 2	
*D. triphylla*: tuber		TPC (mg GAE/100 g FW): 13 ± 1	
*D. japonica*: bulbil	MeOH (80%): fr CF/EA/BuOH/H_2_O	TPC: 2.2 ± 0.1: 11.5 ± 0.4/33.9 ± 1.8/3.9 ± 0.1/2.4 ± 0.1	[89]
*D. batatas*: bulbil		TPC: 3.9 ± 0.2: 19.6 ± 0.8/39.1 ± 2.2/7.4 ± 0.4/5.8 ± 0.2	
*D. hirtiflora*: tubers	Successively DCM/EA/MeOH	TPC (mg GAE/g): 0.25 ± 0.01/8.9 ± 0.69/10.1 ± 0.35 TFC (mg QE/g): ND/24.2 ± 0.43/28.1 ± 0.35	[67]
*D. dumetorum*		TPC (mg GAE/g): 1.75 ± 0.02/0.81 ± 0.003/1.04 ± 0.02 TFC (mg QE/g): 8.58 ± 0.14/12.4 ± 0.43/9.6 ± 0.21	
*D. bulbifera*: mauve		TPC (mg GAE/g): 1.91 ± 0.02/14.0 ± 0.41/5.99 ± 0.09 TFC (mg QE/g): 26.1 ± 0.29/74.4 ± 0.41/54.5 ± 0.73	
*D. bulbifera*: yellow		TPC (mg GAE/g): 1.0 ± 0.10/12.6 ± 0.34/0.99 ± 0.01 TFC (mg QE/g): 7.87 ± 1.10/52.0 ± 0.14/29.3 ± 0.02	

Abbreviations: AA—ascorbic acid; AcA—acetic acid; BuOH—*n*-butanol; C—vitamin C; CE—catechin equivalents; CF—chloroform; CGE—cyanidin glucoside equivalents; C-3-Glc—cianidina-3-galactosıd; CTC—carbon tetrachloride; B—vitamin B; DCM—dichloromethane; dw—dry weight; E—extract; EA—ethyl acetate; edw—extract dry weight; eq—equivalent; EtOH—ethanol; fr—fraction; FW—fresh weight; GAE—gallic acid equivalents; HX—hexane; MeOH—methanol; NaHSO_3_—Na-bisulfide; pdw—plant dry weight; PE—petrol ether; pfw—plant fresh weight; Q—quercetin; QE—quercetin equivalents; RE—rutin equivalents; TAC—total anthocyanin content; TFC—total flavonoid content; TFL—total flavonol content; TPC—total phenol content; TS—total saponins; TT—total tanin; α-Toc—α-tocopherol.

**Table 4 molecules-27-02530-t004:** Bioactivities of *Dioscorea* species reported by the in vitro cell studies and in vivo animal studies.

Species	Preparation	Study Design	Results	Ref.
*D. alata*: tuber	H_2_O-E in plantain and bitter leaf meal	Rat model, 2.0 g dough meal food, consumed within 25 min.	Blood glucose and GI ↓: the potential to be used as functional foods to alleviate postprandial hyperglycemia	[10]
*D. alata*: tuber	MeOH-E	Antimicrobial activity	Effective against the Gram-positive bacteria *Streptococcus pneumoniae* and fungi *Candida albicans*	[21]
*D. alata*: freeze-dried powder	Containing antho-cyanins	TNBS-inducted colitis mice; DACNs at 20, 40, and 80 mg/kg for 3 days, intra-rectally	The levels of pro-inflammatory cytokines, TNF-α and IFN-γ ↓. May be applied as a potential food supplement in inflammatory bowel disease (IBD) therapy	[95]
*D. alata*: tuber	MeOH-E	Antibacterial activity using disc diffusion assay	Effective against the bacteria *Staphylococcus epidermidis* and Gram-negative bacteria: *Shigella dysenteriae, Shigella flexneri*	[50]
*D. alata*: tuber	MeOH (70%)-E	Mice spleen lymphocytes cells	Anti-inflammatory effect by inhibition of the NO and TNF-α expression.	[133]
*D. alata*: tuber	H_2_O (WSP)	Alloxan-induced hyperglycemic rats; 400 mg/kg bw/day, 4 weeks, orally	Fasting blood glucose gradually decreased	[134]
*D. alata* tuber	EtOH and H_2_O	Doxorubicin-induced cardiac damage mice; daily dose 30 mg/mL for 4 weeks; orally	Regulate NF-kB expression at the transcriptional level; cardiac levels of TBARS, ROS, inflammatory factors, the expression of NF-kB, blood pressure ↑; SOD, GPx activity ↑.	[135]
*D. alata*: purple yam	EtOH (80%)-E; EA fr of peel and flesh	Methylglyoxal-induced HepG2 cells	Extracts strengthened antioxidant defense system	[52]
		Antiglycation activity in vitro	Could inhibit the formation of dicarbonyl compounds in a dose-dependent manner.	
*D. batatas*: peel	EtOH (95%), DDP	LPS-induced RAW 264.7 cell model	2, 7-dihydroxy-4, 6-dimethoxy phenanthrene suppressed LPS-induced expression of cytosolic iNOS and COX-2. Could exert anti-inflammatory activity by suppressing NF-κB signaling pathway.	[136]
*D. batatas*: rhizome	H_2_O	STZ-induced diabetic mice; at 500 or 1000 mg/kg 1/day for 4 weeks	Glucose and leptin, total cholesterol, triglycerides, low-density lipoprotein cholesterol ↓. Expression of antioxidant enzymes, and mitochondrial-induced biogenetic factors in the liver, pancreas, and muscle tissue ↑.	[137]
		Allantoin at 20 or 50 mg/kg/day for 4 weeks; orally		
*D. batatas*: flesh and peel	EtOH (60% and 95% *w*/*w*); H_2_O	EtOH-induced gastric ulcer in mice; a single dose 100 or 200 mg/kg bw, orally	Extracts dissolved in: 5% Tween-80, 10% polyethylene glycol, 10% DMSO, and 10% EtOH saline: inflammatory factors, NO and IL-6, in the serum; COX-2 expression in the gastric tissue ↓	[138]
*D. batatas*: bark	EtOH (BDB)	Anti-inflammatory activity in LPS-induced RAW 264.7 cells	NO production (dose-dependent); iNOS protein induction; regulates inflammation by inhibiting the COX-2 pathway	[139]
*D. batatas*: yam	H_2_O-E; powder	STZ-induced diabetic rats; at 2500 or 1000 mg/kg daily doses for 1-month, orally	Fasting blood glucose and HbAlc ↓; the serum antioxidant activities of tGSH, GSH and SOD ↑; lipid malondialdehyde (MDA), oxidized glutathione (GSSG) ↓	[140]
*D. birmanica*: rhizome	EtOH-E	SNP-induced oxidative stress in liver BNL CL.2 cells	Elevated cell viability in a dose-dependent manner. Can ameliorate oxidative stress	[16]
*D. bulbifera*: powder of bulbils	EtOH and H_2_O-E	Cell culture model; 1–100 μg/mL E	Low cytotoxic effect on the cells	[62]
	CF, EA, H_2_O fr	LPS-induced RAW macrophage 264.7 cells	Mild anti-inflammatory activity	
*D. bulbifera*: leaves	MeOH-E	MCF-7 and MDA-MB-231 breast cancer cells	Cytotoxic effect in cell lines; prompts apoptosis at various stages and a significant decrease in viable cells	[63]
*D. cayennensis*: tubers	Protein conc. 64%	Antibacterial activity	No inhibition of *Salmonella* sp. and *Lysteria monocytogenes*; effective against *E. coli*	[141]
*D. hamiltonii*: tuber	MeOH-E	Antimicrobial activity	Effective against *Streptococcus pneumoniae*	[21]
*D. hamiltonii (D. persimilis)*: herb	MeOH-E	Xylene-induced ear edema damage mice; 2 and 6 g/kg, 5 days, orally	Decreased the level the inflammatory cytokines and reduced oxidative stress	[29]
*D. hemsleyi*: poly-saccharides	Warm and hot H_2_O	Anti-hyperglycemic activity	Effectively inhibits α-amylase, α-glucosidase	[72]
*D. japonica*	EtOH-E	λ-carrageenan-induced paw edema mice; 0.5 and 1.0 g/kg, orally	MDA, NO, TNF-α ↓ after 5th hour; 1.0 g/kg decreased the developments of carr-induced paw edema after 5th hour	[142]
		Acute toxicity, at 10 g/kg	No toxicity observed	
		LPS-induced RAW 264.7 cells	No effects on viability; suppressed LPS-induced production of NO, TNF-α, expression of iNOS and COX-2	
*D. japonica*: yam tubers	EtOH and H_2_O-E	Doxorubicin-induced cardiac damage in mice; a daily dose of 30 mg/mL (*w*/*v*) for 4 weeks; orally	Might regulate NF-kB expression at the transcriptional level; cardiac levels of TBARS, ROS, inflammatory factors, expression of NF-kB; blood pressure ↓; SOD, GPX activity ↑	[135]
*Monascus*-fermented D. (red mold D. (RMD)) root	EtOH (95% *w*/*w*) dissolved in mineral oil	DMBA-induced hamster buccal pouch carcinogenesis; RMD extracts (50, 100, 200 mg/kg bw, paint for 14 weeks on days alternate to DMBA painting)	Anti-inflammatory and antioxidative activity. Inhibit the pro-inflammatory cytokines TNF-R, IL-1β, IL-6, and IFN-γ, which in turn, leads to oxidative stress.	[143]
*D. nipponica*	EtOH (70%)-E	Anti-osteosarcoma activity	Induced apoptosis in human osteosarcoma cells line U2OS	[144]
*D. nipponica*: rhizome	Saponins	MSU-inducted gouty arthritis mice; TS at 100, 300, 900 mg/kg every 24 h for 7 days, orally	TS (dioscin, protodioscin, pseudo protodioscin) might restore production of pro-inflammatory cytokines TNF-α, pro-interleukins IL-1β and IL-8 to the normal conditions, regulating antioxidant capacities and NALP3 inflammasome.	[145]
*D. nipponica*: rhizome	EtOH (80%)-E and diosgenin in 1% carboxyl methyl-cellulose	ISO-induced myocardial ischemia model in rats; diosgenin at 20, 40, 80 mg/kg for 3 days. 500 mg/kg for 3 days, orally, after ISO injection	Diosgenin protects the myocardium against ischemic insult through increasing enzymatic and nonenzymatic antioxidant levels; decreasing oxidative stress damage; SOD, CAT, GPx activity ↑; lipid peroxidation ↓ Confirms hypothesis that intestinal bacteria produce diosgenin from *D. nipponica* extract.	[24]
	Dry precipate of saponins	ISO-induced myocardial ischemia model in rats; 150 and 300 mg/kg for 3 days, orally, both before and after ISO injection	SOD, CAT, GPx, total antioxidant capacity (T-AOC) activity ↑; can protect the myocardium against ischemic insult	[146]
*D. nipponica*: rhizome	Dry precipitate of saponins	Rat model; single dose of 160 mg/kg intragastrically	Diosgenin was one of the main metabolites found in plasma and feces. The extract can play an essential role in cardioprotective efficacy.	[146]
*D. nipponica*	Crude drug with saponins	Potassium oxonate-induced hyperuricemic mice; 60, 300, 600 mg/kg every 24 h for 6 days, before induction	Total saponins (dioscin, protodioscin, pseudo protodioscin) from RDN had uricosuric effect and could enhance urate excretion and reduce the serum urate levels	[14]
*D. nipponica*: rhizome	MeOH-E	Antibacterial activity	Effective against *Bacillus subtilis, Staphylococcus aureus, Proteus vulgaris, Salmonella typhimurium*	[45]
*D. opposita*: yam	Fresh-cut, MeOH → yellow powder	^•^OH-induced DNA damage	Can protect against DNA damage (IC_50_ 0.098 ± 0.032 mg/mL); provides a theoretical basis for the application of YP in food and drug industry.	[70]
*D. oppositifolia*	MeOH-E	Xylene-induced ear edema damage mice; 2 and 6 g/kg, for 5 days; orally	The inflammatory cytokines, TNF-α, IL-6; oxidative stress ↓	[29]
*D. oppositifolia*: tubers	MeOH-E	Antimicrobial activity	Effective against *Klebsiella pneumoniae, Shigella dysenteriae, Candida albicans, Candida tropicalis*	[21]
*D. opposita*: Chinese yam	Cold-soaking extract (CYCSE)	HCT-induced rat; CYCSE 60 mg/kg and 80 mg/kg for 10 days H_2_O_2_-induced Leydig cells (TM3)	Can stimulate the NO/cGMP pathway and protect against induced erectile dysfunction; may protect testis morphology, increase TM3 cell proliferation and stimulate testosterone secretion. Suppressed TGF-β1 in injured cells. Can protect against damage from the oxidative stress response.	[147]
*D. panthaica*: rhizome	Precipate of saponins DP-E	ISO-induced myocardial ischemia model in rats; 150 and 300 mg/kg, 3 days; orally	SOD, CAT GPx, total antioxidant capacity (T-AOC) activity ↑; can protect the myocardium against ischemic insult	[146]
*D. purpurea*: tuber	EtOH and H2O	Doxorubicin-induced cardiac damage in mice; a daily dose of 30 mg/mL (*w*/*v*) for 4 weeks; orally	SOD and GPx activity ↑; might regulate NF-kB expression at the transcriptional level; blood pressure, the cardiac levels of TBARS, ROS, and inflammatory factors, the expression of NF kappa B↓.	[135]
*D. pentaphylla*: tuber	MeOH	Antibacterial activity using disc diffusion assay	Effective against *Streptococcus mutans*, *Streptococcus pyogenes*, *Vibrio cholerae*, *Shigella flexneri*, *Salmonella typhi.*	[77]
*D. pubera* Blume	MeOH	Antimicrobial activity	Effective against *Streptococcus pneumoniae*, *Klebsiella pneumoniae, Escherichia coli, Shigella dysenteriae*, *Candida albicans, Candida tropicalis*	[21]
*D. wallichii*	MeOH	Antimicrobial activity	Effective against *Klebsiella pneumoniae, Shigella dysenteriae* and fungus *Candida tropicalis*	[21]
*D. zingiberensis*: rhizome	Total steroid saponins (TSS)	Adjuvant-induced arthritis (AIA) rat; 50, 100, and 200 mg/kg 1/day, every 3 days, respectively, from day 0 to day 28, orally	The levels of pro-inflammatory cytokines IL-1, IL-1β, IL-6, IL-10, and TNF-α ↓; suppressed production of oxidant stress makers: NO, MDA; could protect an injured ankle joint from further deterioration.	[19]
		LPS-induced RAW 264.7 macrophage cells	TSS suppresses NF-κB activation by inhibiting the phosphorylation of p65 and IκBα	
*D. villosa*: leaf	MeOH	Mouse fibroblast L929 cell line	No cytotoxic effect. Scratch assay: expression of Collagen-1; induction of migration of fibroblasts to the wound site ↑.	[104]

Abbreviations: AA—ascorbic acid; AIA—adjuvant-induced arthritis; BDB—bark of *D. batatas* DECNE; cGMP—cyclic 3′,5′-monophosphate; COX-2—cyclooxygenase-2; CYCSE—Chinese yam cold-soaking extract; DACNs—anthocyanins; DP-E—*D. panthaica* extract; DMBA—7,12-dimethylbenz-[a]anthracene; DMSO—*Dimethyl sulfoxide*; E—*extract*; EA—ethyl acetate; EtOH—ethanol; GI—glycemic index; GSH—reduced glutathione; H_2_O_2_—hydrogen peroxide; HbAlc—glycated hemoglobin; HCT—hydrocortisone; IBD—inflammatory bowel disease; IFN-γ—interferon-gamma; (IL)-8—interleukin; iNOS—inducible nitric oxide synthase; ISO— isoprenaline; LPS—lipopolysaccharide; MDA—malondialdehyde; MeOH—methanol; NF kappa B—nuclear factor kappa B; NF-κB—nuclear factor-κB; NO—nitric oxide; Nrf2—nuclear factor-erythroid 2-related factor 2; PGE2—prostaglandin E2; RDN—rhizoma *D. nipponica*; RMD—red mold dioscorea; ROS—reactive oxygen species; SNP—sodium nitroprusside; SOD—superoxide dismutase; STZ—streptozotocin; TBARS—thiobarbituric acid-reacting substances; TGF-β1—transforming growth factor-β1; tGSH—total glutathione; TNBS—trinitrobenzenesulfonic acid; TNF-a—tumor necrosis factor-a; TS—total saponins; TSS—total steroid saponins; YP—yellow powder.

**Table 5 molecules-27-02530-t005:** Phytochemicals in *Dioscorea* spp.

Species: Plant Part	Solvents	Compounds	Ref.
*D. alata*: tubers	MeOH	Vitamins (mg/g): male: C, 13.49 ± 3.64; B1, 1.14 ± 0.16; B2, 1.75 ± 0.26; female: C, 18.26 ± 1.37; B1, 1.26 ± 0.11; B2, 1.87 ± 0.2	[21]
*D. alata*: tubers	EtOH (50%) with 0.1% HCl	Anthocyanins: alatanin C (cyanidin 3-(6-sinapoyl gentiobioside); cyanidin-3-diglucoside; cyanidin-3,5-diglucoside; alatanins B, C; alatanin E, D, F isomers	[93]
*D. alata* (purple yam): dried tubers	EtOH (80%)	Phenolic acids (mg/100 g mdw): galic 0.482 ± 0.057; 4-hydroxy benzoic 0.192 ± 0.0024; syringic 0.899 ± 0.0022; sinapic 0.202 ± 0.0501; chlorogenic 0.451 ± 0.0038; ferulic 0.089 ± 0.0005. Flavonoids (mg/100 g mdw): quercetin 0.687 ± 0.0030, apigenin 0.210 ± 0.0041; kaempferol 9.219 ± 0.0043	[94]
*D. glabra*: tubers		mg/100 gm: C, 23.49 ± 0.0413; H_2_O-soluble B, 0.036 to 4.159	[46]
*D. alata* (purple yam): freeze-dried tubers	MeOH (50%) with 0.1% HCl	Anthocyanins (% peak area): cyanidin-3,5-diglucoside (31.22); cyanidin-3-diglucoside-5-celery glycosides (28.77); delphinidin-3-glucose-5-rutinoside (16.36); delphinidin-3-glucoside (12.18), delphinidin-3,5-diglucoside (11.31)	[95]
*D. alata*: freeze-dried tubers	MeOH (70%)	Phenolic acids (mg/g dw): galic 29.34; 4-hydroxy benzoic 6.48; syringic 2.94; p-coumaric 2.53; myricetin 42.39	[133]
*D. alata*: tubers	MeOH	Myricetin, gallic acid, ellagic acid, vanillic acid, syringic acid, epicatechin, vanillin, *p*-coumaric acid, *trans*-cinnamic acid and kaempferol.	[50]
*D. batatas*: freeze-dried flesh and peel of tubers	EtOH (95%)	Phenanthrenes (mg/100 g dw): peel: 2,7-dihydroxy-4,6-dimethoxy 47.35 ± 0.25; 6,7-dihydroxy-2,4-dimethoxy 29.29 ± 0.08; 6-hydroxy-2,4,7-trimethoxy (batatasin I) 35.85 ± 0.12	[54]
*D. batatas*: yam	Thermally treated meals	Vitamins (mg/100 g): E, 8.3; C, 3.5; B_1_, 2.1, B_2_, 0.03	[84]
*D. bulbifera*: flesh and peel of bulbils/tubers	MeOH	Phenolic acids (µg/g dw). Flesh: gallic 1.69 ± 0.13, isovanillic 1.02 ± 0.06, protocatechuic 0.15 ± 0.01. Peel: gallic 2.30 ± 0.20, isovanillic 0.18 ± 0.02, protocatechuic 0.10 ± 0.02 Flavonoids (μg/g dw). Flesh: catechin: 46.1 ± 0.75, quercetin 0.08 ± 0.01. Peel: catechin: 8.50 ± 1.01, quercetin 0.27 ± 0.05 Ascorbic acid (µg/g dw). Flesh: 26.4 ± 0.51. Peel: 25.03 ± 3.82	[107]
	EA	Phenolic acids (µg/g dw). Flesh: gallic 0.32 ± 0.14, isovanillic 1.71 ± 0.05, protocatechuic 0.13 ± 0.007. Peel: gallic 0.27 ± 0.03, isovanillic 0.22 ± 0.09, protocatechuic 0.20 ± 0.07 Flavonoids (μg/g dw). Flesh: catechin 108.3 ± 0.69, quercetin 0.99 ± 0.05. Peel: catechin 23.1 ± 0.22, quercetin 1.36 ± 0.16 Ascorbic acid (µg/g dw). Flesh: 4.52 ± 1.18. Peel: 2.8 ± 0.09	
*D. japonica*: leaves	DE extract fraction	Total triterpenoids (including esters) (mg/g d.w.): Tokyo 734.71; Kanagawa 716.55	[169]
*D. caucasica*: leaves	DE	Total of triterpenoids (including esters) (mg/g dw): 1492.56	[169]
*D. hispida*: leaves	DE extract fraction	Total of triterpenoids (including esters) (mg/g dw): 704.11	[169]
*D. quinquelobata*: leaves	DE extract fraction	Total of triterpenoids (including esters) (mg/g dw): 467.29	[169]
*D. purpurea*: leaves	DE extract fraction	Total of triterpenoids (including esters) (mg/g dw): 628.54	[169]
*D. nipponica*: leaves	DE extract fraction	Total of triterpenoids (including esters) (mg/g dw): 837.83	[169]
*D. hamiltonii*: dried powder of tubers	MeOH	Vitamins (mg/g): male: ascorbic acid 10.31 ± 2.75, thiamine 1.15 ± 0.09, riboflavin 0.82 ± 0.07; female: ascorbic acid 12.7 ± 3.64, thiamine 1.03 ± 0.16, riboflavin 0.98 ± 0.12	[21]
*D. hamiltonii* (syn *D. persimilis*): dry herb powder	MeOH; HPLC	Phenolic acids (µg/g d.w): gallic 6.24 ± 0.07; protocatechuic 0.65 ± 0.02; chlorogenic 0.93 ± 0.03; syringic 26.26 ± 0.42; p-coumaric 0.96 ± 0.05 Flavonoids (µg/g dw): catechin 17.69 ± 0.03; rutin 7.07 ± 0.22; quercetol 6.8 ± 0.17; kaempferol 5.92 ± 0.13. Saponin content (µg/g dw): protogracillin 60.21 ± 1.04; dioscin 18.21 ± 0.54; diosgenin 25.00 ± 0.08; trillin 107.08 ± 1.12	[29]
*D. hirtiflora*: flesh/peel	MeOH	Phenolic acids (µg/g dw): gallic 0.34 ± 0.00/0.73 ± 0.35, isovanillic 0.28 ± 0.04/0.30 ± 0.06, protocatechuic 0.13 ± 0.02/0.13 ± 0.03. Flavonoids (μg/g dw): catechin 6.91 ± 0.21/4.00 ± 0.23, quercetin 0.47 ± 0.14/0.28 ± 0.009 Ascorbic acid (µg/g d.w): 5.88 ± 0.57/12.0 ± 0.61	[107]
	EA	Phenolic acids (µg/g dw): gallic 0.19 ± 0.03/0.24 ± 0.04, isovanillic 0.88 ± 0.03/0.93 ± 0.09, protocatechuic 0.42 ± 0.02/0.20 ± 0.02. Flavonoids (μg/g d.w): catechin 23.7 ± 0.42/5.19 ± 0.50, quercetin 0.42 ± 0.09/1.57 ± 0.26. Ascorbic acid (µg/g d.w): 4.26 ± 0.39/2.72 ± 0.22	
*D. nipponica*: rhizomes	NADES containing 30% H_2_O	Steroidal saponins (%): protodioscin 79.90, protogracillin 68.12, pseudoprotodioscin 67.27, pseudoprotogracillin 74.8	[170]
*D. nipponica*: freeze-dried rhizomes	EtOH (70%)	Saponins (mg/g): protodioscin 159.983 ± 0.064; protogracillin 4.250 ± 0.024; pseudoprotodioscin 13.821 ± 0.037; dioscin 22.999 ± 0.121	[144]
*D. opposita*: rhizome	Hot H_2_O	Phenolic acids (μg/g): gallic 3.56 ± 0.13; chlorogenic 6.77 ± 0.06; vanillic 8.49 ± 0.36; syringic 2.95 ± 0.14; p-coumaric 16.90 ± 0.17. Flavonoids (μg/g): epicatechin 7.37 ± 0.24; phlorizin 18.90 ± 0.48	[75]
	EtOH (80%)	Phenolic acids (μg/g): gallic 3.10 ± 0.4; chlorogenic 7.92 ± 0.42; vanillic 12.59 ± 0.51; syringic 6.78 ± 0.46; p-coumaric 16.39 ± 0.37 Flavonoids (μg/g): rutin 9.48 ± 0.40; epicatechin 28.39 ± 0.57; phlorizin 20.29 ± 0.34	
*D. oppositifolia*.: stems, leaves	AC (50%) → DCM	Norsesquiterpenoids: dioscopposin A, dioscopposin B	[151]
*D. oppositifolia* (*syn* D. *opposita*) (Chinese yam): herb	MeOH	Phenolic acids (μg/g dw): gallic 3.67 ± 0.10; protocatechuic 0.69 ± 0.02; chlorogenic 1.20 ± 0.04; vanillic 2.08 ± 0.05; syringic 37.35 ± 0.49; p-coumaric 1.65 ± 0.04. Flavonoids (μg/g dw): rutin 11.98 ± 0.16; quercetol 27.76 ± 0.12; kaempferol 18.65 ± 0.08; catechin 4.45 ± 0.07. Saponins (μg/g dw): protogracillin 154.45 ± 2.56; dioscin 23.64 ± 0.27; diosgenin 26.02 ± 0.05; trillin 77.61 ± 0.10	[29]
*D. quinquelobata*: rhizomes	EtOH (70%)	Steroidal saponins (mg/g): protodioscin 3.496 ± 0.018, protogracillin 5.945 ± 0.020, pseudoprotodioscin ND., dioscin 10.002 ± 0.051, gracillin 9.011 ± 0.098	[144]
*D. pentaphylla*: leaves	MeOH	Gallic acid, rutin, quercetin	[76]
*D. polystachya*; tubers	MeOH (70%)	Dehydroepiandrosterone, allantoin, 5-hydroxy-7-methoxyflavanone, arnebinone, dioscin, protodioscin	[171]
*D. pubera*: tubers	MeOH	Vitamin content (mg/g): male: ascorbic acid 14.29 ± 2.38, thiamine 0.85 ± 0.07, riboflavin 1.02 ± 0.08; female: ascorbic acid 15.88 ± 1.37, thiamine 0.99 ± 0.11, riboflavin 0.94 ± 0.14	[21]
*D. septemloba*: rhizomes	EtOH (75%)	Phenanthropyran: dioscorone B, phenanthrene: 2,2′,6,6′-tetramethoxy-4,4′7,7′-tetrahydroxy-1,1′-biphenanthrenes	[172]
*D. septemloba*: rhizomes	EtOH (70%)	Steroidal saponins (mg/g): protodioscin 8.959 ± 0.014, protogracillin 9.902 ± 0.061, pseudoprotodioscin ND, dioscin 9.822 ± 0.014, gracillin 7.123 ± 0.031	[144]
*D. wallichii*: dried powder	MeOH	Vitamins (mg/g): male: ascorbic acid 9.52 ± 2.38, thiamine 1.25 ± 0.13, riboflavin 1.18 ± 0.1; female: ascorbic acid 12.7 ± 1.4, thiamine 1.11 ± 0.12, riboflavin 1.13 ± 0.24	[21]
*D. bulbifera:* rhizomes	Purified from EtOH (80%) E with EA, Sephadex LH-20, and ODS	C22 ω-hydroxy fatty acid, 3-hydroxy-5-methoxybenzoic acid, various phenanthrene derivatives and flavonoids	[173]
*D. trifida*: yam tubers		Pelargonidin, cyanidin, peonidin glycosides and other derivatives	[113]
*D. opposita*		Aromatic benzyl compounds, dihydrostilbenes, phenanthrenes: diarylheptanoids, apigenin	[80]
*D. opposita*: aerial parts	EtOH	6,7-dihydroxy-2-methoxy-1,4-phenanthrenedione; chrysoeriol 4′-*O*-â-*D*-glucopyranoside, chrysoeriol 7-*O*-â-*D*-glucopyranoside, alternanthin, daucosterol	[174]
*D. communis*		Herorensol, 2,3,4-trimethoxy-7,8-methylenedioxyphenanthrene, 2,4-dimethoxy-7,8-methylendioxy-3-phenanthrenol, chrysotoxene, 2,4,8-trimethoxy-3,7-phenanthrenediol, orchinol, and lusianthridin 7	[31]
*D. schimperiana*: yellow-fleshed/yellow with red dot/red-fleshed	MeOH (60%)	µg/100 g: α-Toc 538.66/275.11/554.86; lutein: 18.06/16.35/18.22; zeaxanthin: 11.62/6.70/11.68; β-kryptoxanthin: 2.37/6.12/2.39; β-carotene: 212.23/197.04/212.98; β-carotene: 560.94/462.30/562.91; lycopene: 0.83/0.84/0.84; Pro-vit A carotenoids: 787.15/672.16/789.95	[82]
*D. batatas*: thermally treated		(mg/100 g dw) Chlorophyll a/b: 0.43 ± 0.01/0.75 ± 0.02; lycopene 0.30 ± 0.00; phytic acid 1.04 ± 0.42	[56]

Abbreviations: AC—acetone; C—vitamin C; DCM—dichloromethane; DE—diethyl ether; dw—dry weight; EA—ethyl acetate; EtOH—ethanol; FW—fresh weight; HCl—hydrochloric acid; MeOH—methanol; NADES—natural deep eutectic solvent; ND—not detected; α-Toc—α-tocopherol.

## Data Availability

Not applicable.

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
