# Peer review of "Dioscorea spp.: Comprehensive Review of Antioxidant Properties and Their Relation to Phytochemicals and Health Benefits"

_molecules, 2022, doi:10.3390/molecules27082530_

Round 1

Reviewer 1 Report

Regarding to the manuscript entitled "Comprehensive review of antioxidant properties and their relation to phytochemicals and health benefits" I have some comments and suggestions, hope lead to improve its quality:

Abstract

- please add an overall finding of the review

- the keywords must be revised, add more correlated words

Introduction

L48-79: I would suggest allocating a distinct section for traditional uses of the species

L91-99: this paragraph is vague to me, I do not understand the aim of that, discussing phytochemicals? bioactivities? please re-write

L150: please make the genus/species names italic throughout the text

in Table 2 (main text s well), it can be better to unify the IC50/EC50 units to easier comparing bioactivities by readers, although it is already explained by authors "L174-175: Therefore, when the reader would like to know more details, it is suggested to upload the full article", but not clear; moreover changing of the units is not a problematic issue in most of cases, e.g. μg/mL to mg/mL

- in Table 2, what is RP? reducing power of what? please describe

- I would suggest separating the related studies of TPC/TFC (Table 3 and text) from antioxidant properties with distinct headings, however the findings can be discussed after those parts

- in some parts of manuscript (L395,396: Dough meal with 40% yam strongly inhibited α-amylase and α-glucosidase, which resulted in the lowest postprandial glycemic index [2].) and section 3, the other bioactivities of the genus are described, however as authors mentioned several published reviews have gathered those information, moreover it is not in line with the aim of present study; besides, section 3 describes other biological effects in a very un-classified manner, on the other hand it is not easy to find out the desired potencies 

- please add chemical structures of the major phytoconstituents

Conclusion

- it is expected to summarize overall findings of the review, however I would recommend mentioning the highest antioxidant properties belong to which species, which phytoconstituents, which plant part, which extract, these information can conduct further studies on the genus

Reviewer 2 Report

The manuscript offers a methodical review of articles devoted to studying the Dioscorea spp antioxidant potential, the presence of bioactives and their relations to health benefits, processing techniques, in vitro and in vivo assays, etc. A very large number of articles published has been examined and analyzed and the information is gathered and presented accordingly. The organization of the manuscript is good.

One recommendation: The purpose of a review article is not only to present the state-of-the-art but also, to outline some perspectives and mark new avenues to be pursued.

From the data reported in Table 5, (Section 4. Phytochemicals of Dioscorea) it can be concluded that the main solvents applied are organic like methanol, ethanol, DE, etc. The authors mention that there are just few papers devoted to the application of more "modern extraction methods" like microwave and ultra sound assisted extractions of Dioscorea.

In view of the above, a brief new paragraph should be included stating that there is a clear-cut need to investigate the implementation of predominantly mild, advanced processes and techniques (e.g. extraction with compressed fluids - SCE, PLE, etc), that will allow obtaining extracts comprising bioactives with a wide spectrum of applications without damaging one or more of the antioxidants particularly those that are heat sensitive.

Reviewer 3 Report

The manuscript "Dioscorea spp.: Comprehensive review of antioxidant properties and their relation to phytochemicals and health benefits" is well written and covers a large literature about the plant genera.

The major focus is on antioxidant activities in vitro despite the action in animal models or studies in population that consume the plant as food.  Phytochemical analysis was shown as a topic and analysis of the identity of compounds as well.

I think that this manuscript did not fall into the scope proposed to the  Special Issue Antioxidants from Natural Sources: Separation and Characterization as a paper to Natural Products Chemistry.

Most information inside of the manuscript did not describe the "Separation" or "Characterization" of plant constituents and maybe is not suitable to be published on Molecule.

A major review must be made to get focus on the Special Issue.

Round 2

Reviewer 1 Report

Concerning the manuscript entitled "Comprehensive review of antioxidant properties and their relation to phytochemicals and health benefits", authors have tried to revised the requested/requested items, but unfortunately most are not corrected. Taking into account that the purpose for peer reviewing is improving the quality of manuscript, thus I can not accept some of the rationales for not revising them, for instance the request for adding structures, although some are added but they are inserted like copy/paste a picture, please draw with e.g. ChemDraw. the following answer is not acceptable "The structures of the majority listed in Table 5 phytochemicals are easily available from the internet", the structures for every compounds can be found on internet, so what is the aim of the review manuscript? is not gathering the information regarding the topic?

- In Table 2, "reducing power" must be described in total definition, reducing power of what? which factor/agent?

- regarding the conclusion, the reason is also not acceptable. The overall results of the review is not written in the abstract nor in the conclusion. Highly recommended adding those data.

Reviewer 3 Report

After revision, the manuscript did not was improved. I think that this manuscript did not fall into the scope proposed to the  Special Issue Antioxidants from Natural Sources: Separation and Characterization as a paper to Natural Products Chemistry.

Most information inside of the manuscript did not describe the "Separation" or "Characterization" of plant constituents and maybe is not suitable to be published on Molecule.
